# Simple Process-Led Algorithms for Simulating Habitats (SPLASH v.1.0): Robust Indices of Radiation, Evapotranspiration and Plant-Available Moisture

Tyler W. Davis[1,*], I. Colin Prentice[1,2,3,4], Benjamin D. Stocker[1], Rebecca T. Thomas[1], Rhys J. Whitley[2,4], Han Wang[2,3], Bradley J. Evans[4,5], Angela V. Gallego-Sala[6], Martin T. Sykes[7], and Wolfgang Cramer[8]

[1]AXA Chair of Biosphere and Climate Impacts, Grand Challenges in Ecosystems and the Environment and Grantham Institute - Climate Change and the Environment, Department of Life Sciences, Imperial College London, Silwood Park Campus, Ascot, United Kingdom
[2]Department of Biological Sciences, Macquarie University, North Ryde, New South Wales, Australia
[3]State Key Laboratory of Soil Erosion and Dryland Farming on the Loess Plateau, College of Forestry, Northwest Agriculture & Forestry University, Yangling 712100, China
[4]Terrestrial Ecosystem Research Network (TERN) Ecosystem Modelling and Scaling Infrastructure (eMAST), Sydney, New South Wales, Australia
[5]Faculty of Agriculture and Environment, Department of Environmental Sciences, The University of Sydney, Sydney, New South Wales, Australia
[6]Department of Geography, University of Exeter, Exeter, Devon, United Kingdom
[7]Department of Physical Geography and Ecosystem Science, Lund University, Lund, Sweden
[8]Mediterranean Institute of marine and terrestrial Biodiversity and Ecology (IMBE), Aix Marseille University, CNRS, IRD, Avignon University, Aix-en-Provence, France
[*]now at: United States Department of Agriculture-Agricultural Research Service, Robert W. Holley Center for Agriculture and Health, Ithaca, United States

*Correspondence to:* I. Colin Prentice (c.prentice@imperial.ac.uk)

**Abstract.** Bioclimatic indices for use in studies of ecosystem function, species distribution, and vegetation dynamics under changing climate scenarios depend on estimates of surface fluxes and other quantities, such as radiation, evapotranspiration and soil moisture, for which direct observations are sparse. These quantities can be derived indirectly from meteorological variables, such as near-surface air temperature, precipitation and cloudiness. Here we present a consolidated set of Simple Process-Led Algorithms for Simulating Habitats (SPLASH) allowing robust approximations of key quantities at ecologically relevant time scales. We specify equations, derivations, simplifications and assumptions for the estimation of daily and monthly quantities of top-of-the-atmosphere solar radiation, net surface radiation, photosynthetic photon flux density, evapotranspiration (potential, equilibrium and actual), condensation, soil moisture, and runoff, based on analysis of their relationship to fundamental climatic drivers. The climatic drivers include a minimum of three meteorological inputs: precipitation, air temperature, and fraction of bright sunshine hours. Indices, such as the moisture index, the climatic water deficit, and the Priestley-Taylor coefficient, are also defined. The SPLASH code is transcribed in C++, FORTRAN, Python, and R. One year of results are presented at the local and global scales to exemplify the spatiotemporal patterns of daily and monthly model outputs along with comparisons to other model results.

# 1 Introduction

Despite the existence of dense networks of meteorological monitoring stations around the world, plant ecophysiology and biogeography suffer from a lack of globally distributed observational data, especially those central to the estimation of ecosystem-level photosynthesis, including photosynthetic photon flux density and soil moisture. To overcome this deficiency, we present Simple Process-Led Algorithms for Simulating Habitats (SPLASH) for generating driving datasets for ecological and land-surface models (e.g., monthly carbon and water fluxes or seasonal plant functional trait distributions) from more readily available meteorological observations.

SPLASH is a continuation of the STASH (STAtic SHell) model, which was originally developed for modeling the climatic controls on plant species distributions at a regional scale (Sykes and Prentice, 1995, 1996; Sykes et al., 1996). The intention of STASH was to provide bioclimatic indices, reflecting the environment experienced by plants more closely than either standard summary variables such as mean annual temperature, or such constructions as 'mean precipitation of the warmest quarter,' while requiring only standard meteorological data as input. A key component of STASH was a simple, physically-based soil moisture accounting scheme, first developed by Cramer and Prentice (1988), which has been used *inter alia* in the original, highly cited BIOME model (Prentice et al., 1992); the general forest succession model (FORSKA) described by Prentice et al. (1993); and the Simple Diagnostic Biosphere Model (Knorr and Heimann, 1995). Despite the subsequent development of more complex Dynamic Global Vegetation Models (Cramer et al., 2001; Sitch et al., 2003; Woodward and Lomas, 2004; Quillet et al., 2010; Prentice and Cowling, 2013; Fisher et al., 2014) and Land Surface Models, the relatively simple algorithms in STASH continue to have many applications, including to new areas such as the distribution of plant functional traits (Harrison et al., 2010; Meng et al., 2015), assessment of climate-change impacts on specific biomes (Gallego-Sala and Prentice, 2012), large-scale water resources assessments (e.g. Ukkola et al., 2015) and simple first-principles modeling of primary production (Wang et al., 2014). The continuing utility of these algorithms owes much to their robustness, which in turn depends on the implicit assumption that vegetation functions predictably—so that, for example, evapotranspiration occurs at a potential rate under well-watered conditions, and is reduced as soil water is drawn down. STASH is thus unsuitable to answer questions like the effect of imposed vegetation changes on runoff, or modeling vegetation-atmosphere feedbacks. Much more complex models that dynamically couple soil, vegetation and atmospheric boundary layer processes exist for such applications; however, their complexity brings a burden in terms of lack of robustness and, potentially, large inter-model differences (Prentice et al., 2014).

Despite their long history of use, no single publication documents the algorithms of the STASH model. This work aims to fill that gap to allow for the continued development and use of these algorithms. As the new incarnation of STASH, SPLASH provides the same physically-based soil moisture accounting scheme with updated and corrected analytical expressions for the calculation of daily radiation, evapotranspiration, and soil moisture. Included in this documentation are the equation derivations, variable definitions, and information regarding model assumptions and limitations. One notable improvement is that we have discontinued the approximation of constant angular velocity in the orbit of Earth around the Sun. This version is thus suitable

for palaeoclimate applications, whereby orbital precession (as well as changes in obliquity and eccentricity) influences the seasonal distribution of insolation. SPLASH also includes explicit consideration of elevation effects on biophysical quantities.

Key model outputs include daily insolation (incoming solar radiation at the top of the atmosphere) and net surface radiation ($H_o$ and $H_N$, respectively); daily photosynthetic photon flux density ($Q_n$); daily condensation, soil moisture and runoff ($C_n$, $W_n$, and $RO$); and daily equilibrium, potential and actual evapotranspiration ($E_n^q$, $E_n^p$, and $E_n^a$). Unlike the STASH model, SPLASH explicitly distinguishes potential and equilibrium evapotranspiration, recognizing that under well-watered conditions the excess of the former over the latter is a requirement for foliage to be cooler than the surrounding air, as has long been observed under high environmental temperatures (e.g. Linacre, 1967).

Input values of latitude, $\phi$ (rad), elevation, $z$ (m), mean daily near-surface air temperature, $T_{air}$ (°C), and fractional hours of bright sunshine, $S_f$ (unitless), are used for calculating the daily quantities of net radiation and evapotranspiration. Daily precipitation, $P_n$ (mm d$^{-1}$), is used for updating daily soil moisture. $T_{air}$ and $P_n$ may be derived from various sources, including the freely available daily-averaged air temperature and precipitation reanalysis data from the Water and Global Change (WATCH) program's meteorological forcing data set (Weedon et al., 2014). Meteorological variables are also available in the Climatic Research Unit (CRU) gridded monthly time series datasets (Harris et al., 2014), which may be downscaled to daily quantities by means of quasi-daily methods (e.g., linear interpolation). Cloud cover fraction, for example the simulated quantities given in the CRU TS3.21 dataset, may be used to approximate $S_f$. Penman's one-complement approximation based on the cloudiness fraction is regarded here as a sufficient estimate of $S_f$ (Penman, 1948). The piecewise linear method of Hulme et al. (1995)—an adaptation of the Doorenbos-Pruitt estimation procedure (Doorenbos and Pruitt, 1977)—as used in the development of the CRU cloudiness climatology (New et al., 1999) gives similar results.

We present SPLASH comprehensively re-coded in a modular framework to be readable, understandable and reproducible. To facilitate varied application requirements (including computational speed), four versions of the code (C++, FORTRAN, Python, and R) are available in an online repository (see Code Availability). The algorithms as presented here focus on application to individual site locations, but a natural extension is towards spatially distributed grid-based datasets.

In line with the intention of the original STASH algorithms, we also present bioclimatic indices at the monthly and annual timescales to exemplify the analytical applications of the SPLASH model outputs.

## 2 Methodology

The implementation of the soil-moisture accounting scheme follows the steps outlined by Cramer and Prentice (1988), where daily soil moisture, $W_n$ (mm), is calculated based on the previous day's moisture content, $W_{n-1}$, incremented by daily precipitation, $P_n$ (mm d$^{-1}$), and condensation, $C_n$ (mm d$^{-1}$), and reduced by daily actual evapotranspiration, $E_n^a$ (mm d$^{-1}$) and runoff, $RO$ (mm):

$$W_n = W_{n-1} + P_n + C_n - E_n^a - RO, \tag{1}$$

where $P_n$ is a model input, $C_n$ is estimated based on the daily negative net radiation, $E_n^a$ is the analytical integral of the minimum of the instantaneous evaporative supply and demand rates over a single day, and $RO$ is the amount of soil moisture

in excess of the holding capacity. An initial condition of $W_n$ is assumed between zero and the maximum soil moisture capacity, $W_m$ (mm), for a given location and is equilibrated over an entire year by successive model iterations (i.e., model spin-up). Under steady-state conditions, the SPLASH model preserves the water balance, such that $\sum (P_n + C_n) = \sum (E_n^a + RO)$.

To solve the simple 'bucket model' represented by Eq. 1, the following steps are taken at the daily timescale: calculate the radiation terms, estimate the condensation, estimate the evaporative supply, estimate the evaporative demand, calculate the actual evapotranspiration, and update the daily soil moisture. Daily quantities may be aggregated into monthly and annual totals and used in moisture index calculations.

## 2.1  Radiation

### 2.1.1  Top-of-the-atmosphere solar radiation

The calculation of $C_n$ and $E_n^a$ begin with modeling the extraterrestrial solar radiation flux, $I_o$ (W m$^{-2}$). The equation for $I_o$ may be expressed as the product of three terms (Duffie and Beckman, 2013):

$$I_o = I_{sc}\, d_r\, \cos\theta_z, \tag{2}$$

where $I_{sc}$ (W m$^{-2}$) is the solar constant, $d_r$ (unitless) is the distance factor, and $\cos\theta_z$ (unitless) is the inclination factor. Values for $I_{sc}$ may be found in the literature (e.g., Thekaekara and Drummond, 1971; Willson, 1997; Dewitte et al., 2004; Fröhlich, 2006; Kopp and Lean, 2011); a constant for $I_{sc}$ is given in Table 2.

The distance factor, $d_r$, accounts for additional variability in $I_o$ that reaches the Earth. This variability is due to the relative change in distance between Earth and the Sun caused by the eccentricity of Earth's elliptical orbit, $e$ (unitless), and is calculated as (Berger et al., 1993):

$$d_r = \left( \frac{1 + e\,\cos\nu}{1 - e^2} \right)^2, \tag{3}$$

where $\nu$ (rad) is Earth's true anomaly. True anomaly is the measure of Earth's location around the Sun relative to its position when it is closest to the Sun (perihelion).

The last term, $\cos\theta_z$, attenuates $I_o$ to account for the Sun's height above the horizon (measured relative to the zenith angle, $\theta_z$), accounting for the off-vertical tilt of Earth's rotational axis, $\varepsilon$ (i.e., obliquity). The inclination factor is calculated as (Duffie and Beckman, 2013):

$$\cos\theta_z = \sin\delta\,\sin\phi + \cos\delta\,\cos\phi\,\cos h, \tag{4}$$

where $\phi$ (rad) is the latitude, $\delta$ (rad) is the declination angle, and $h$ (rad) is the hour angle, measuring the angular displacement of the Sun east or west of solar noon ($-\pi \le h \le \pi$). Declination is the angle between Earth's equator and the Sun at solar noon ($h = 0$), varying from $+\varepsilon$ at the June solstice to $-\varepsilon$ at the December solstice; the changing declination is responsible for the change in seasons. For the purposes of ecological modeling, $\delta$ may be assumed constant throughout a single day. See e.g. Woolf (1968) for the precise geometric equation representing $\delta$:

$$\delta = \arcsin\left(\sin\lambda\,\sin\varepsilon\right), \tag{5}$$

where $\lambda$ (rad) is Earth's true longitude (i.e., the heliocentric longitude relative to Earth's position at the vernal equinox) and $\varepsilon$ (rad) is obliquity (i.e., the slowly varying tilt of Earth's axis). Several other methods are widely used for the estimation of $\delta$ for a given day of the year (e.g., Cooper, 1969; Spencer, 1971; Swift, 1976) but are not recommended because they do not account for the change in Earth's orbital velocity with respect to the distance between Earth and the Sun, while Eq. 5 does. The relationship between true longitude, $\lambda$, and true anomaly, $\nu$, is by the angle of the perihelion with respect to the vernal equinox, $\widetilde{\omega}$ (rad) (Berger, 1978):

$$\nu = \lambda - \widetilde{\omega}. \tag{6}$$

While the three orbital parameters (i.e., $e$, $\varepsilon$, and $\widetilde{\omega}$) exhibit long-term variability (on the order of tens of thousands of years), they may be treated as constants for a given epoch (e.g., $e = 0.0167$, $\varepsilon = 23.44°$, and $\widetilde{\omega} = 283.0°$ for 2000 CE), or they may be calculated using the methods of Berger (1978) or Berger and Loutre (1991) for palaeoclimate studies. Berger (1978) presents a simple algorithm to estimate $\lambda$ for a given day of the year (see Appendix A).

The daily top-of-the-atmosphere solar radiation, $H_o$ ($\mathrm{J\,m^{-2}}$), may be calculated as twice the integral of $I_o$ measured between solar noon and the sunset angle, $h_s$, assuming that all angles related to Earth on its orbit are constant over a whole day:

$$H_o = \int_{day} I_o = 2 \int_{h=0}^{h_s} I_o = \frac{86\,400}{\pi} I_{sc}\, d_r \, (r_u\, h_s + r_v\, \sin h_s), \tag{7}$$

where $r_u = (\sin\delta\,\sin\phi)$ and $r_v = (\cos\delta\,\cos\phi)$, both unitless.

The sunset angle can be calculated as the hour angle when the solar radiation flux reaches the horizon (i.e., when $I_o = 0$) and can found by substituting Eq. 4 into Eq. 2, setting $I_o$ equal to zero, and solving for $h$:

$$h_s = \arccos\left(-\frac{r_u}{r_v}\right). \tag{8}$$

To account for the undefined negative fluxes produced by Eq. 2 for $h \geq h_s$ and $h \leq -h_s$, $I_o$ should be set equal to zero during these nighttime hours. To account for the occurrences of polar day (i.e., no sunset) and polar night (i.e., no sunrise), $h_s$ should be limited to $\pi$ when $r_u/r_v \geq 1$ and zero when $r_u/r_v \leq -1$.

### 2.1.2 Net surface radiation

The net surface radiation, $H_N$ ($\mathrm{J\,m^{-2}}$), is the integral of the net surface radiation flux received at the land surface, $I_N$ ($\mathrm{W\,m^{-2}}$), which is classically defined as the difference between the net incoming shortwave radiation flux, $I_{SW}$ ($\mathrm{W\,m^{-2}}$) and the net outgoing longwave radiation flux, $I_{LW}$ ($\mathrm{W\,m^{-2}}$):

$$I_N = I_{SW} - I_{LW}. \tag{9}$$

The calculation of $I_{SW}$ is based on the reduction in $I_o$ due to atmospheric transmittivity, $\tau$ (unitless), and surface shortwave albedo, $\beta_{sw}$ (unitless):

$$I_{SW} = (1 - \beta_{sw})\,\tau\,I_o. \tag{10}$$

A constant value for $\beta_{sw}$ is given in Table 2. Atmospheric transmittivity may be expressed as a function of elevation (to account for attenuation caused by the mass of the atmosphere) and cloudiness (to account for atmospheric turbidity). At higher elevations, there is less atmosphere through which shortwave radiation must travel before reaching the surface. To account for this, Allen (1996) presents an equation based on the regression of Beer's radiation extinction function at elevations below 3000 m with an average sun angle of 45°, which can be expressed as:

$$\tau = \tau_o \left(1 + 2.67 \times 10^{-5} z\right), \tag{11}$$

where $z$ (m) is the elevation above mean sea level and $\tau_o$ (unitless) is the mean sea-level transmittivity, which can be approximated by the Ångstrom-Prescott formula:

$$\tau_o = c + d\, S_f, \tag{12}$$

where $c$ and $d$ are empirical constants (unitless) and $S_f$ is the fraction of daily bright sunshine hours ($0 \leq S_f \leq 1$). Values for $c$ and $d$ are given in Table 2.

The calculation of $I_{LW}$ is based on the difference between outgoing and incoming longwave radiation fluxes attenuated by the presence of clouds, which may be empirically estimated by (Linacre, 1968):

$$I_{LW} = [b + (1 - b)\, S_f]\,(A - T_{air}), \tag{13}$$

where $A$ and $b$ are empirical constants and $T_{air}$ (°C) is the mean near-surface air temperature. The outgoing longwave radiation flux used to derive Eq. 13 assumes a constant ground emissivity, which is accurate under well-watered conditions. The incoming longwave radiation flux is modeled based on clear-sky formulae derived by Linacre (1968). Values for $A$ and $b$ are given in Table 2.

$H_N$ can be decomposed into its net positive, $H_N^+$ (J m$^{-2}$), and net negative, $H_N^-$ (J m$^{-2}$), components (i.e., $H_N = H_N^+ + H_N^-$). Assuming $I_{LW}$ is constant throughout the day and making substitutions for $I_o$ into Eq. 10, an expression for $H_N^+$ may be derived as twice the integral of $I_N$ between solar noon (i.e., $h = 0$) and the net surface radiation flux cross-over hour angle, $h_n$ (rad):

$$H_N^+ = 2 \int_{h=0}^{h_n} I_N = \frac{86\,400}{\pi} \left[(r_w\, r_u - I_{LW})\, h_n + r_w\, r_v\, \sin h_n\right], \tag{14}$$

where $r_w = (1 - \beta_{sw})\, \tau\, I_{sc}\, d_r$ (W m$^{-2}$).

Here, $h_n$ is the hour angle when $I_{SW}$ equals $I_{LW}$ and can be found by setting $I_N = 0$ in Eq. 9 and solving for $h$, following the same substitutions as used for $h_s$ in Eq. 8, and may be expressed as:

$$h_n = \arccos\left(\frac{I_{LW} - r_w\, r_u}{r_w\, r_v}\right). \tag{15}$$

To account for the occurrences when the net surface radiation flux does not cross the zero datum, $h_n$ should be limited to $\pi$ when $(I_{LW} - r_w\, r_u)/(r_w\, r_v) \leq -1$ (i.e., net surface radiation flux is always positive) and zero when $(I_{LW} - r_w\, r_u)/(r_w\, r_v) \geq 1$ (i.e., net surface radiation flux is always negative).

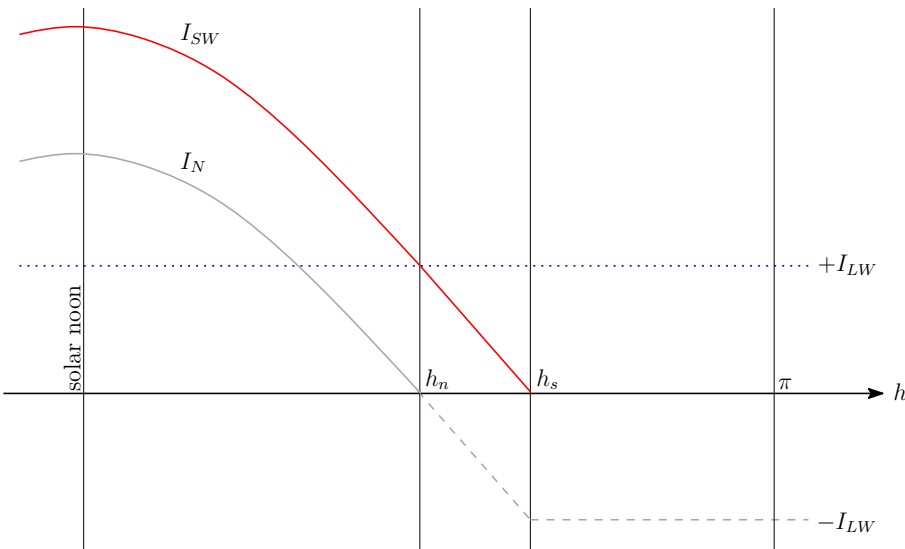

**Figure 1.** Example of the net radiation flux curve between the hours of solar noon (i.e., $h = 0$) and solar midnight (i.e., $h = \pi$). The $I_N$ curve is equal to the difference between the net incoming shortwave radiation flux, $I_{SW}$ (solid red line), and the net outgoing longwave radiation flux, $I_{LW}$ (dotted blue line). Positive $I_N$, shown decreasing from solar noon to zero at the cross-over hour angle, $h_n$, is denoted with a solid gray line, while negative $I_N$, shown decreasing from zero to $-I_{LW}$ between $h_n$ and the sunset hour angle, $h_s$, and constant between $h_s$ and solar midnight, is denoted with a dashed gray line. The solid black horizontal line marks the datum of zero radiation.

Complementary to $H_N^+$, $H_N^-$ may be calculated as twice the integral of $I_N$ between $h_n$ and solar midnight, defined by the piecewise function of $I_N$ between $h_n$ and $h_s$ and $-I_{LW}$ between $h_s$ and solar midnight (i.e., $h = \pi$), given as (note that $H_N^-$ is a negative quantity):

$$H_N^- = 2 \left( \int_{h_n}^{h_s} I_N - \int_{h_s}^{\pi} I_{LW} \right) = \frac{86\,400}{\pi} \left[ r_w\, r_v\, (\sin h_s - \sin h_n) + r_w\, r_u\, (h_s - h_n) - I_{LW}\, (\pi - h_n) \right]. \tag{16}$$

5    Figure 1 shows an example of a half-day $I_N$ curve used in the integrals defined in Eqns. 14 and 16. $I_N$, which is at its peak at solar noon, crosses zero at $h_n$ and reaches a minimum at $h_s$. After sunset (i.e., $h > h_s$), when $I_{SW}$ is zero, $I_N$ is equal to $-I_{LW}$. $H_N^+$ is represented as twice the integral under the positive net radiation curve (solid gray line), above the zero line (solid black line), and between the vertical lines of solar noon and $h_n$. $H_N^-$ is represented as twice the integral below the zero line and above the negative net radiation curve (dashed gray line).

10    **2.1.3    Photosynthetically active radiation**

The daily photosynthetically active radiation in units of photon flux density, $Q_n$ $(\mathrm{mol\,m^{-2}\,d^{-1}})$, is calculated based on the number of quanta received (moles of photons) within the visible light spectrum, which also corresponds to the action spectrum

of photosynthesis (Monteith and Unsworth, 1990):

$$Q_n = 1 \times 10^{-6} \, \text{fFEC} \, (1 - \beta_{vis}) \, \tau \, H_o, \tag{17}$$

where $\beta_{vis}$ (unitless) is the visible light albedo and fFEC ($\mu$mol J$^{-1}$) is the flux-to-energy conversion factor (Ge et al., 2011). This factor takes into account both the portion of visible light within the total solar spectrum, approximately 50% (Stanhill and Fuchs, 1977), and the mean number of quanta in the visible light energy band, approximately 4.6 $\mu$mol J$^{-1}$ (McCree, 1972). The $1 \times 10^{-6}$ converts the units of $Q_n$ from $\mu$mol m$^{-2}$ d$^{-1}$ to mol m$^{-2}$ d$^{-1}$. Values for $\beta_{vis}$ and fFEC are given in Table 2.

## 2.2 Condensation

The daily condensation, $C_n$, may be expressed as the water-equivalent of the absolute value of negative net radiation, $H_N^-$:

$$C_n = 1 \times 10^3 \, E_{con} \, |H_N^-|, \tag{18}$$

where $E_{con}$ (m$^3$ J$^{-1}$) is the water-to-energy conversion factor that relates the energy released or required for a unit volume of water to evaporate or condense at a given temperature and pressure, based on a simplification of the Priestley and Taylor (1972) potential evapotranspiration for a horizontally uniform saturated surface, which may be expressed as:

$$E_{con} = \frac{s}{L_v \, \rho_w \, (s + \gamma)}, \tag{19}$$

where $s$ (Pa K$^{-1}$) is the slope of the saturation vapor pressure-temperature curve, $L_v$ (J kg$^{-1}$) is the latent heat of vaporization of water, $\rho_w$ (kg m$^{-3}$) is the density of water, and $\gamma$ (Pa K$^{-1}$) is the psychrometric constant. Standard values may be assumed for certain parameters (e.g., $L_v = 2.5 \times 10^6$ J kg$^{-1}$; $\rho_w = 1 \times 10^3$ kg m$^{-3}$; $\gamma = 65$ Pa K$^{-1}$); however, equations for the temperature dependence of $s$ and $L_v$ (e.g., Allen et al., 1998; Henderson-Sellers, 1984) and the temperature and pressure dependence of $\rho_w$ and $\gamma$ (e.g., Kell, 1975; Chen et al., 1977; Allen et al., 1998; Tsilingiris, 2008) are available (see Appendix B).

The barometric formula may be used to estimate the atmospheric pressure, $P_{atm}$ (Pa), at a given elevation, $z$ (m), when observations are not available. Assuming a linear decrease in temperature with height, which is a reasonable approximation within the troposphere (i.e., for $z < 1.10 \times 10^4$ m), the following equation may be used (Berberan-Santos et al., 1997):

$$P_{atm} = P_o \left(1 - \frac{L \, z}{T_o}\right)^{\frac{g \, M_a}{R \, L}}, \tag{20}$$

where $P_o$ (Pa) is the base pressure, $T_o$ (K) is the base temperature, $z$ (m) is the elevation above mean sea level, $L$ (K m$^{-1}$) is the mean adiabatic lapse rate of the troposphere, $g$ (m s$^{-2}$) is the standard gravity, $M_a$ (kg mol$^{-1}$) is the molecular weight of dry air, and $R$ (J mol$^{-1}$ K$^{-1}$) is the universal gas constant. Values for the constants used in Eq. 20 are given in Table 2.

## 2.3 Evaporative Supply

The evaporative supply rate, $S_w$ $(\mathrm{mm\,h^{-1}})$ is assumed to be constant over the day and can be estimated based on a linear proportion of the previous day's soil moisture, $W_{n-1}$ (Federer, 1982):

$$S_w = S_c \frac{W_{n-1}}{W_m}, \tag{21}$$

where $S_c$ $(\mathrm{mm\,h^{-1}})$ is the supply rate constant (i.e., maximum rate of evaporation) and $W_m$ (mm) is the maximum soil moisture capacity. Constant values for $S_c$ and $W_m$ are given in Table 2.

## 2.4 Evaporative Demand

The evaporative demand rate, $D_p$ $(\mathrm{mm\,h^{-1}})$, is set equal to the potential evapotranspiration rate, $E^p$ $(\mathrm{mm\,h^{-1}})$, as defined by Priestley and Taylor (1972). $E^p$ usually exceeds the equilibrium evapotranspiration rate, $E^q$ $(\mathrm{mm\,h^{-1}})$, due to the entrainment of dry air in the convective boundary layer above an evaporating surface (Raupach, 2000, 2001). $E^p$ is related to $E^q$ by the Priestley-Taylor coefficient, which may be defined as one plus an entrainment factor, $\omega$ (Lhomme, 1997):

$$D_p = E^p = (1+\omega)\,E^q. \tag{22}$$

The constant value used for $\omega$ is given in Table 2. The calculation of $E^q$ is based on the energy-water equivalence of $I_N$, ignoring the soil heat flux, (Lhomme, 1997):

$$E^q = 3.6 \times 10^6\,E_{con}\,I_N, \tag{23}$$

where $3.6 \times 10^6$ converts the units of $E^q$ from $\mathrm{m\,s^{-1}}$ to $\mathrm{mm\,h^{-1}}$. Note that $E^q$ is defined only for positive values (i.e., $E^q = 0$ for $I_N < 0$). The Priestley-Taylor potential evapotranspiration is preferred in this context to the general Penman-Monteith equation for actual evapotranspiration (Penman, 1948; Monteith, 1965), which requires knowledge of stomatal and aerodynamic conductances, or to any of the 'reference evapotranspiration' formulae (Allen et al., 1998) that specifically relate to agricultural crops.

Daily equilibrium evapotranspiration, $E_n^q$ $(\mathrm{mm\,d^{-1}})$, is based on the integration of Eq. 23 for values of positive $I_N$, or simply the energy-water equivalence of $H_N^+$:

$$E_n^q = 1 \times 10^3\,E_{con}\,H_N^+, \tag{24}$$

where $1 \times 10^3$ converts $E_n^q$ from $\mathrm{m\,d^{-1}}$ to $\mathrm{mm\,d^{-1}}$.

The daily demand, which is equal to the daily potential evapotranspiration, $E_n^p$ $(\mathrm{mm\,d^{-1}})$, may be calculated from $E_n^q$, as in Eq. 22:

$$E_n^p = (1+\omega)\,E_n^q. \tag{25}$$

## 2.5 Actual Evapotranspiration

The calculation of daily actual evapotranspiration, $E_n^a$ (mm d$^{-1}$), is based on the daily integration of the actual evapotranspiration rate, $E^a$ (mm h$^{-1}$), which may be defined as the minimum of the evaporative supply and demand rates (Federer, 1982):

$$E^a = \min\left(S_w, D_p\right), \tag{26}$$

where $S_w$ (mm h$^{-1}$) is the evaporative supply rate, defined in Eq. 21, and $D_p$ (mm h$^{-1}$) is the evaporative demand rate, defined in Eq. 22.

The analytical solution to $E_n^a$ may be expressed analogous to the methodology used for solving $H_o$ and $H_N$ and is defined as twice the integral of $E^a$ between solar noon and $h_n$, which comprises two curves: $S_w$ for $0 \le h \le h_i$ and $D_p$ for $h_i \le h \le h_n$, where $h_i$ (rad) is the hour angle corresponding to the intersection of $S_w$ and $D_p$ (i.e., when $S_w = D_p$):

$$E_n^a = 2 \int_{h=0}^{h_n} E^a = 2 \left( \int_0^{h_i} S_w + \int_{h_i}^{h_n} D_p \right), \tag{27a}$$

which may be expressed as:

$$E_n^a = \frac{24}{\pi} \left[ S_w\, h_i + r_x\, r_w\, r_v\, (\sin h_n - \sin h_i) + (r_x\, r_w\, r_u - r_x\, I_{LW})\, (h_n - h_i) \right], \tag{27b}$$

where $r_x = 3.6 \times 10^6\, (1 + \omega)\, E_{con}$ (mm m$^2$ W$^{-1}$ h$^{-1}$). The intersection hour angle, $h_i$, is defined by setting Eq. 21 equal to Eq. 22 and solving for $h$:

$$h_i = \arccos\left( \frac{S_w}{r_x\, r_w\, r_v} + \frac{I_{LW}}{r_w\, r_v} - \frac{r_u}{r_v} \right). \tag{28}$$

To account for the occurrences when supply is in excess of demand during the entire day, $h_i$ should be limited to zero when $\cos h_i \ge 1$. For occurrences when supply limits demand during the entire day, $h_i$ should be limited to $\pi$ when $\cos h_i \le -1$.

Figure 2 shows an example of the half-day evaporative supply and demand rate curves. $D_p$ (dashed red line) is at a maximum at solar noon and decreases down to zero at $h_n$, while $S_w$ (dotted blue line) is constant throughout the day. The point where
$S_w$ equals $D_p$ is denoted by the vertical bar at $h_i$. $E^a$ (solid gray line), limited by supply during most of the day, follows the $S_w$ line between solar noon and $h_i$. During the time between $h_i$ and $h_n$, $E^a$ no longer limited by supply, follows the $D_p$ curve. After $h_n$, both $D_p$ and $E^a$ are zero. $E_n^a$ is represented by twice the area above the zero line (horizontal solid black line), below the $E^a$ line, and between the vertical bars of solar noon and $h_n$.

## 2.6 Runoff

The calculation of daily runoff, $RO$, is based on the excess of daily soil moisture without runoff compared to the holding capacity, $W_m$, and is given by:

$$RO = \max\left(0, W_n{}^* - W_m\right), \tag{29}$$

where $W_n{}^*$ (mm) is the daily soil moisture without runoff (i.e., Eq. 1 where $RO = 0$).

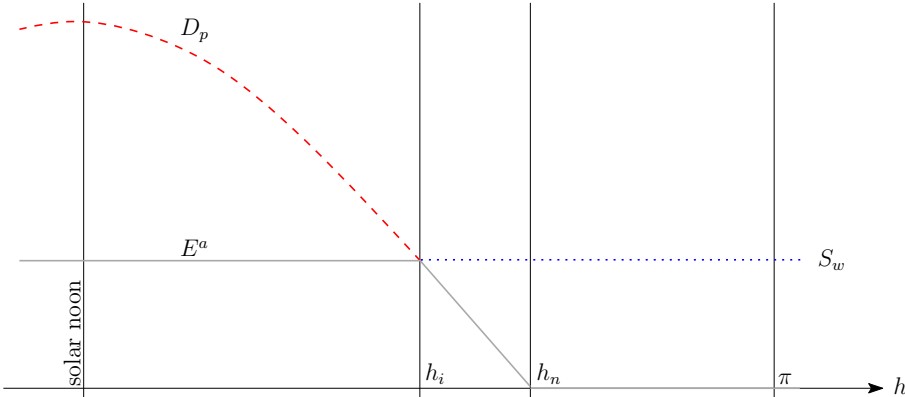

**Figure 2.** Example of actual evapotranspiration curve between the hours of solar noon (i.e., $h = 0$) and solar midnight (i.e. $h = \pi$). The evaporative demand, $D_p$ (dashed red line), is at a maximum at solar noon and zero at the cross-over hour angle, $h_n$. The evaporative supply, $S_w$ (dotted blue line), is constant throughout the day. The point where supply is equal to demand denotes the intersection hour angle, $h_i$. Actual evapotranspiration (solid gray line) is defined as the minimum of $S_w$ and $D_p$ throughout the day.

## 2.7   Soil Moisture

With analytical expressions for $C_n$, $E_n^a$ and $RO$ (i.e., Eqns. 18, 27b and 29, respectively), $W_n$ may now be calculated by Eq. 1. Once $W_n$ is calculated, the following limits are checked:

$$0 \leq W_n \leq W_m. \tag{30}$$

The calculation of $RO$ in Eq. 29 should prevent $W_n$ from being greater than $W_m$, thus satisfying the upper limit of Eq. 30.

The limiting effect of $S_w$ on $E_n^a$, through Eqns. 27 and 28, should, in most cases, prevent $W_n$ from falling below zero and satisfy the lower limit of Eq. 30; however, due to the assumption that $S_w$ is constant throughout the day, there is the possibility that $E_n^a + RO$ may exceed $W_{n-1} + P_n + C_n$, resulting in negative $W_n$. In these rare cases, in order to maintain the mass balance of the bucket model presented in Eq. 1, $E_n^a$ is reduced by an amount equal to the magnitude of the negative soil moisture.

## 3   Bioclimatic Indices

One application of the SPLASH model is estimating the surface fluxes required for the calculation of bioclimatic indices. Typically described at longer time scales (e.g., monthly or annually), the daily SPLASH fluxes can be integrated to monthly and annual totals:

$$X_{m,a} = \sum_{d=1}^{N_{m,a}} X_d, \tag{31}$$

where $X$ is a model output parameter at a given day ($X_d$), month ($X_m$), or year ($X_a$) and $N$ is the total number of days to sum

over for a given month ($N_m$) or a given year ($N_a$).

The following sections describe three common bioclimatic indices.

## 3.1 Moisture Index

There exists a long history that includes several variants of the moisture index, $MI$, also commonly referred to as the aridity index, $AI$, or moisture ratio, $MR$ (Thornthwaite, 1948; Budyko, 1961). A current definition describes $MI$ as the ratio of annual precipitation to annual potential evapotranspiration (Middleton and Thomas, 1997), given as:

$$MI = \frac{P_a}{E_a^p}, \tag{32}$$

where $P_a$ ($\mathrm{mm\,a^{-1}}$) is the annual precipitation and $E_a^p$ ($\mathrm{mm\,a^{-1}}$) is the annual potential evapotranspiration as calculated by Eq. 31; $P_a$ and $E_a^p$ may be substituted with their multi-year means (i.e., $\bar{P}_a$ and $\bar{E}_a^p$) if available. Values less than one are indicative of annual moisture deficit.

## 3.2 Climatic Water Deficit

The climatic water deficit, $\Delta E$, defined as the difference between the evaporative demand (i.e., potential evapotranspiration) and the actual evapotranspiration, has been shown to be a biologically meaningful measure of climate as it pertains to both the magnitude and length of drought stress experienced by plants (Stephenson, 1998). At the monthly timescale, this index is calculated as:

$$\Delta E_m = E_m^p - E_m^a, \tag{33}$$

where $\Delta E_m$ ($\mathrm{mm\,mo^{-1}}$) is the monthly climatic water deficit, $E_m^p$ ($\mathrm{mm\,mo^{-1}}$) is the monthly potential evapotranspiration and $E_m^a$ ($\mathrm{mm\,mo^{-1}}$) is the monthly actual evapotranspiration. $E_m^p$ and $E_m^a$ are the monthly totals of $E_n^p$ and $E_n^a$, respectively, calculated by Eq. 31. Values of $\Delta E$ may also be computed at the annual timescale.

## 3.3 Priestley-Taylor Coefficient

The Priestley-Taylor coefficient, $\alpha$, is the ratio of actual evapotranspiration to equilibrium evapotranspiration, which represents the fraction of plant-available surface moisture (Priestley and Taylor, 1972; Sykes et al., 1996; Gallego-Sala et al., 2010). At the monthly timescale, this is defined as:

$$\alpha_m = \frac{E_m^a}{E_m^q}, \tag{34}$$

where $\alpha_m$ is the monthly Priestley-Taylor coefficient, $E_m^a$ is the monthly actual evapotranspiration and $E_m^q$ ($\mathrm{mm\,mo^{-1}}$) is the monthly equilibrium evapotranspiration. Due to the entrainment factor, $\alpha_m$ may vary between zero (i.e., no moisture) and $1+\omega$ (i.e., unlimited moisture). Values of $\alpha$ may also be computed at the annual timescale.

## 4 Results

The methodology described in Sect. 2 was translated into computer application code (C++, FORTRAN, Python and R). The following sections describe the year-long SPLASH simulation results (2000 CE) at the local and global scales along with comparisons with other model results.

### 4.1 Local Temporal Trends and Bioclimatic Indices

The SPLASH model was run at six locations across North America (see Fig. 3), representing six distinct climate regions across latitudinal and elevational gradients. Ten years (i.e., 1991–2000) of monthly CRU TS3.23 data (i.e., precipitation, air temperature, and cloudiness fraction) were extracted from the $0.5° \times 0.5°$ pixel located over each site. Air temperature and cloudiness fraction were assumed constant and monthly precipitation was divided equally across each day of the month. Fractional sunshine hours were calculated as the one-complement of the cloudiness fraction. Orbital parameters (for paleoclimatology studies) were assumed constant and calculated for the 2000 CE epoch based on the methods of Berger (1978). Model constants were assigned as per Table 2.

The first year of the simulation (i.e., 1991) was iterated (approximately twice) until the daily soil moisture, initialized at zero, reached equilibrium, after which the model was spun-up for eight years (i.e., 1992–1999). The results, shown in Figs. 4 and 5, are for the year 2000. Accompanying the daily SPLASH results in Fig. 4, shown in red, are daily surface fluxes based on the three-layer Variable Infiltration Capacity (VIC) model, extracted from the $1/16°$ pixel over each of the six sites from the datasets provided by Livneh et al. (2015).

Figure 4a shows the daily results for a tundra region over Banff National Park in Alberta, Canada with a mean annual temperature of $-4\,°\mathrm{C}$ and annual precipitation of $986\,\mathrm{mm}$. The SPLASH $H_N$ depicts a bell-shaped curve characteristic for the northern hemisphere. During the spring and summer months, SPLASH $H_N$ is higher than the VIC results, which exhibit a lower $H_N$ during the first half of the year. The SPLASH $W_n$ remains saturated throughout the year at a level between the second and third layers of VIC. SPLASH and VIC $E_n^p$ are similar in magnitude throughout the year with SPLASH $E_n^a$ following $E_n^p$ all year.

Figure 4b shows the daily results for a continental warm summer region over the Adirondack region of New York with a mean annual temperature of $5\,°\mathrm{C}$ and annual precipitation of $1080\,\mathrm{mm}$. There is agreement between the SPLASH and VIC $H_N$ and $E_n^p$ throughout the year. The SPLASH $W_n$ remains saturated throughout most of the year with a dry-down period during mid to late summer and a recovery period during the autumn.

Figure 4c shows the daily results for a temperate region with dry summers over the Bay Area of California with a mean annual temperature of $14\,°\mathrm{C}$ and annual precipitation of $594\,\mathrm{mm}$. During the dry summer months, SPLASH $H_N$ is slightly higher than VIC, during which time the SPLASH $W_n$ is depleted causing moisture-limited $E_n^a$ to occur. Before which, during the winter and early spring, the SPLASH $W_n$ is saturated and $E_n^a$ follows $E_n^p$.

Figure 4d shows the daily results for a hot arid desert region in southwestern Arizona with a mean annual temperature of $23\,°\mathrm{C}$ and annual precipitation of $39\,\mathrm{mm}$. Over the entire year, SPLASH $H_N$ is higher than VIC, with the largest differences

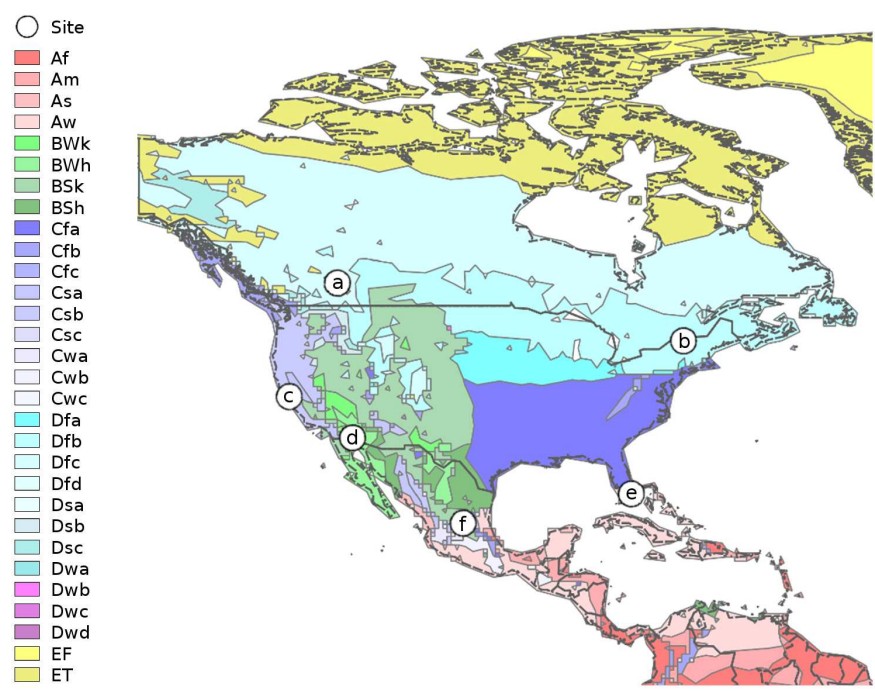

**Figure 3.** Map of Köppen-Geiger climate regions across North America (Kottek et al., 2006). Six locations are selected representing: (a) tundra—ET (51.8° N, 116.5° W, 1383 m.a.s.l.); (b) continental with warm summers—Dfb (44.7° N, 73.8° W, 383 m.a.s.l.); (c) temperate with dry summers—Csb (37.8° N, 122.4° W, 16 m.a.s.l.); (d) hot arid desert—BWh (32.7° N, 114.6° W, 43 m.a.s.l.); (e) equatorial monsoon—Am (26.0° N, 80.3° W, 2 m.a.s.l.); (f) cold arid steppe—BSk (22.2° N, 101.0° W, 1850 m.a.s.l.).

occurring during the summer months. The SPLASH and VIC $W_n$ are both consistently low throughout the year. This water limitation is expressed in the low SPLASH $E_n^a$. During the summer months, SPLASH $E_n^p$ is slightly higher than VIC.

Figure 4e shows the daily results for an equatorial monsoonal region near the southern tip of Florida with a mean annual temperature of 24 °C and annual precipitation of 1500 mm. There is agreement between SPLASH and VIC $H_N$ throughout the year; however, SPLASH $W_n$ is higher than all three layers of VIC except for a few days following a large rain event in

5   October. During the drier winter, there is a slight moisture limitation shown in the SPLASH $E_n^a$. Throughout the year, SPLASH $E_n^p$ is slightly higher than VIC.

Figure 4f shows the daily results for a cold arid steppe region in San Luis Potosí, Mexico with a mean annual temperature of 18 °C and annual precipitation of 346 mm. During the winter, SPLASH $H_N$ is slightly higher than VIC. The SPLASH $W_n$ remains low throughout the year at a level between the first and second layers of VIC. The moisture limitation results in a

10  lower SPLASH $E_n^a$ throughout the year. The SPLASH and VIC $E_n^p$ agree during the year.

Figure 5 shows the SPLASH monthly integrated evapotranspiration results ($E_m^p$ in solid black, $E_m^q$ in dotted black, and $E_m^a$ in dashed red) along with two monthly bioclimatic indices: $\Delta E_m$ and $\alpha_m$. For both the tundra and continental climate sites

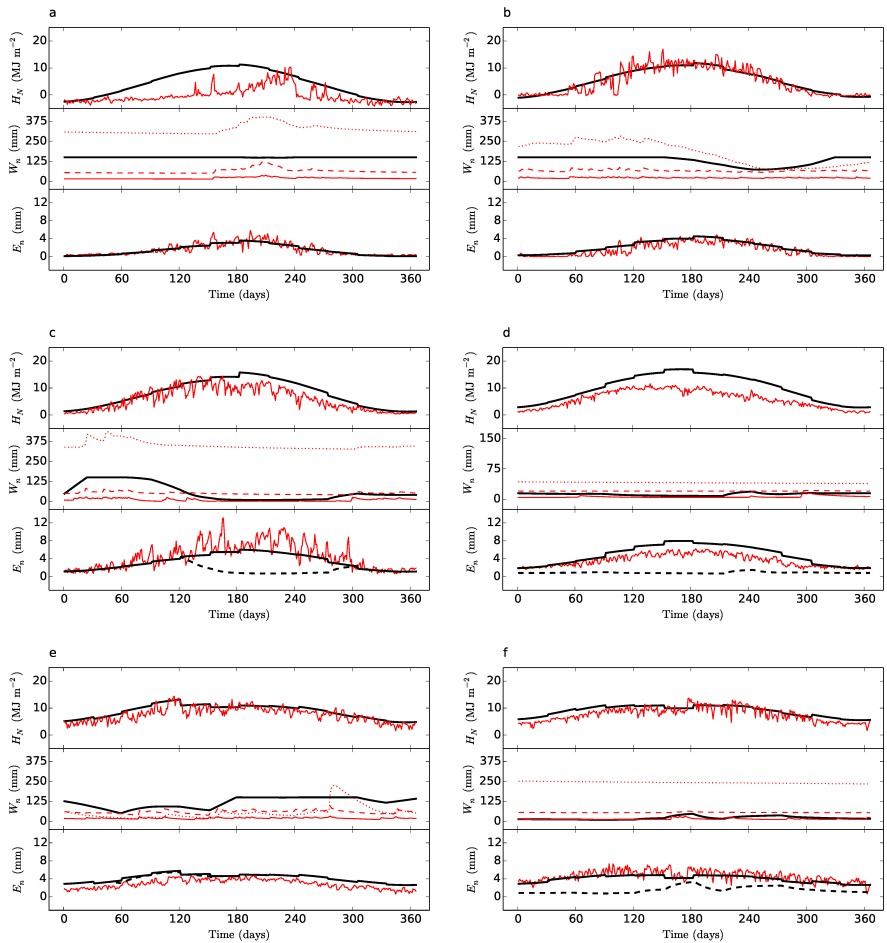

**Figure 4.** Daily simulations of net radiation, $H_N$, soil moisture, $W_n$, and evapotranspiration, $E_n$ for the six climate regions defined in Fig. 3: (a) tundra, (b) continental with warm summers, (c) temperate with dry summers, (d) hot arid desert, (e) equatorial monsoon, and (f) cold arid steppe. Black lines represent SPLASH modeled net radiation, soil moisture, and evapotranspiration (potential in solid black and actual in dashed black). Red lines represent VIC three-layer modeled surface fluxes from Livneh et al. (2015) for net radiation, soil moisture (layer 1 in solid red, layer 2 in dashed red, and layer 3 in dotted red), and potential evapotranspiration. Days of the year are represented along the x-axis. Data are for one year (2000 CE).

(Figs. 5a and 5b, respectively), $E_m^a$ is equivalent to $E_m^p$, which results in constant indices for $\Delta E_m$ (i.e., 0 mm) and $\alpha_m$ (i.e., 1.26). At the annual time scale, $\Delta E_a$ and $\alpha_a$ for these two sites are the same as their monthly values and $MI$ is greater than one, suggesting that these are water available sites.

Figure 5c shows the monthly SPLASH results for the temperate with dry summers region. Similar to the daily results (i.e., Fig. 4c), during the dry summer, $E_m^a$ falls below the $E_m^p$ and $E_m^q$ curves. This results in a positive $\Delta E_m$ and a drop in $\alpha_m$ during the summer months. At the annual time scale, $\Delta E_a$ is 619 mm, which is slightly higher than the annual precipitation

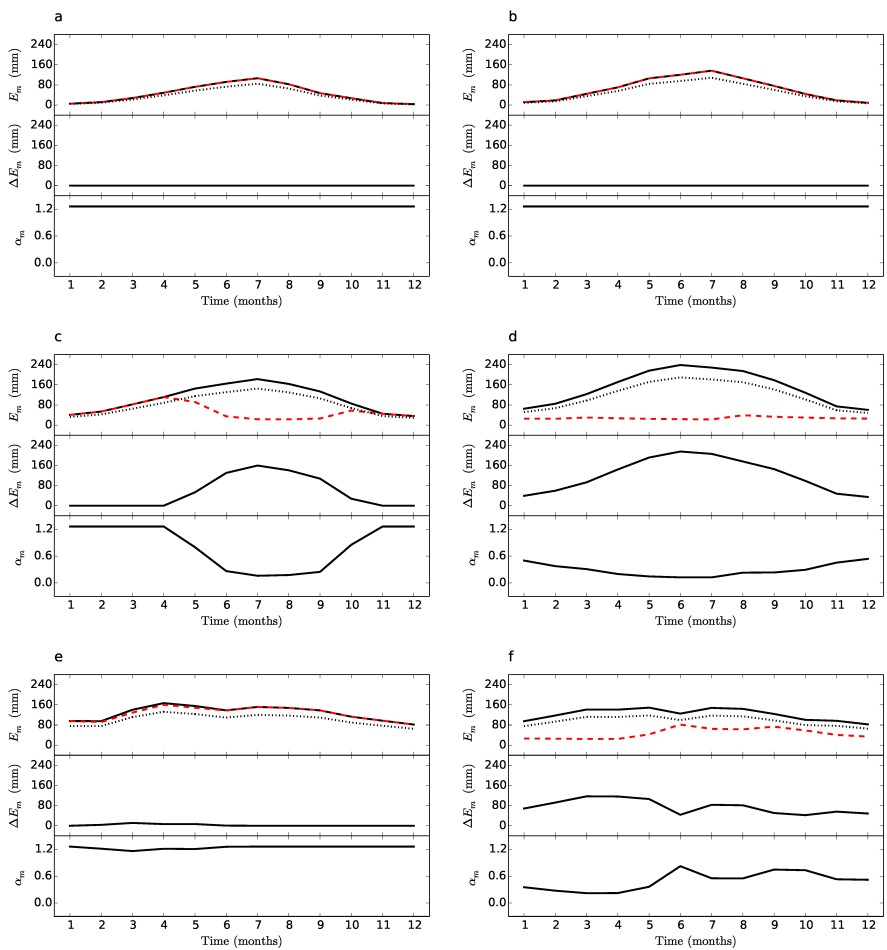

**Figure 5.** Monthly SPLASH results of evapotranspiration, $E_m$ (potential in solid black, actual in dashed red, and equilibrium in dotted black), climatic water deficit, $\Delta E_m$, and Priestley-Taylor coefficient, $\alpha_m$ for the six climate regions defined in Fig. 3: (a) tundra, (b) continental with warm summers, (c) temperate with dry summers, (d) hot arid desert, (e) equatorial monsoon, and (f) cold arid steppe. Months of the year are represented along the x-axis. Results are of one year (2000 CE).

(i.e., 594 mm), and both $\alpha_a$ and $MI$ are less than one (i.e., 0.633 and 0.477, respectively) suggesting that this is a water-limited site.

The hot arid desert region presents a more extreme case as shown in Fig. 5d, where $E_m^a$ is constantly below both $E_m^p$ and $E_m^q$. This results in a positive bell-shaped $\Delta E_m$ curve and a shallow bowl-shaped $\alpha_m$ curve. At the annual time scale, $\Delta E_a$ is 1450 mm, which is significantly higher than the annual precipitation (i.e., 39 mm). Also, both $\alpha_a$ and $MI$ are significantly less than one (i.e., 0.236 and 0.0219, respectively), suggesting that this is a critically water-limited site.

In contrast at the equatorial monsoonal site, shown in Fig. 5e, $E_m^a$ closely follows the $E_m^p$ curve, which results in a nearly zero $\Delta E_m$ and a nearly constant 1.26 $\alpha_m$. At the annual time scale, $\Delta E_a$ is 29 mm, $\alpha_a$ is 1.24, and $MI$ is 0.985, which all suggest that this site is not water limited.

Similar to the hot arid desert, at the high elevation of the cold arid steppe, shown in Fig. 5f, $E_m^a$ is constantly below both $E_m^p$ and $E_m^q$. Unlike the hot arid desert site, the cold arid steppe site is at a lower latitude, which results in a flatter $H_N$ curve (as shown in Fig. 4f) that leads to a more constant $E_m^p$ curve. At the annual time scale, $\Delta E_a$ is 905 mm, which is greater than the annual precipitation (i.e., 346 mm). Both $\alpha_a$ and $MI$ are less than one (i.e., 0.482 and 0.236, respectively), which suggests that this is a water-limited site.

## 4.2 Global Simulation of Spatial Patterns

For the global simulation, $0.5° \times 0.5°$ CRU TS3.23 data were assembled for one year (2000 CE), including monthly precipitation ($\mathrm{mm\,mo^{-1}}$), monthly mean daily air temperature ($°\,\mathrm{C}$), and monthly cloudiness fraction. Monthly precipitation was converted to daily precipitation by dividing the rainfall equally amongst the days in the month. Fractional sunshine hours were calculated based on the one-complement of cloudiness fraction and assumed constant over the month. Mean daily air temperature was also assumed constant over each day of the month. Half-degree land-surface elevation (m above mean sea level) was provided by CRU TS3.22 (Harris et al., 2014). Once again, orbital parameters were assumed constant over the year and calculated for the 2000 CE epoch based on the methods of Berger (1978) and model constants were assigned as per Table 2.

The SPLASH simulations were driven by the data described above, one pixel at a time, starting each pixel with an empty bucket and terminating when a steady-state of soil moisture was reached between the first and last day of the year. Following the spin-up to equilibrate the soil moisture, the model was driven once again to produce daily simulations of net radiation and soil moisture.

Figure 6b and 6e show the SPLASH results of the mean daily net surface radiation flux ($\mathrm{MJ\,m^{-2}}$) for the months of June and December, respectively. For comparison, the Clouds and the Earth's Radiant Energy System (CERES) Energy Balanced and Filled (EBAF) average all-sky surface net total flux for June and December 2000 are plotted in Fig. 6a and 6d, respectively. The CERES EBAF net downward radiative flux was converted from $\mathrm{W\,m^{-2}}$ to $\mathrm{MJ\,m^{-2}}$.

Overall, the SPLASH model produces a reasonable simulation of the latitudinal gradients and seasonal shifts of net surface radiation flux. The differences between SPLASH and CERES EBAF net downward radiative flux are highlighted in Fig. 6c and 6e. Regions in red indicate areas where the SPLASH model results are higher than the CERES EBAF values, while regions in gray indicate areas where the SPLASH model results are lower. The locations where the SPLASH model disagrees with CERES EBAF tend to occur where the well-watered constant surface albedo assumption fails, such as in deserts and at high-latitude ice sheets and tundra.

Figure 7b and 7e show the SPLASH results of the mean daily relative soil moisture (%) for the months of June and December, respectively. An ice sheet was imposed over Greenland (i.e., no soil moisture). For comparison, the National Center for Environmental Prediction (NCEP) Climate Prediction Center (CPC) Version 2 mean soil moisture (van den Dool et al., 2003)

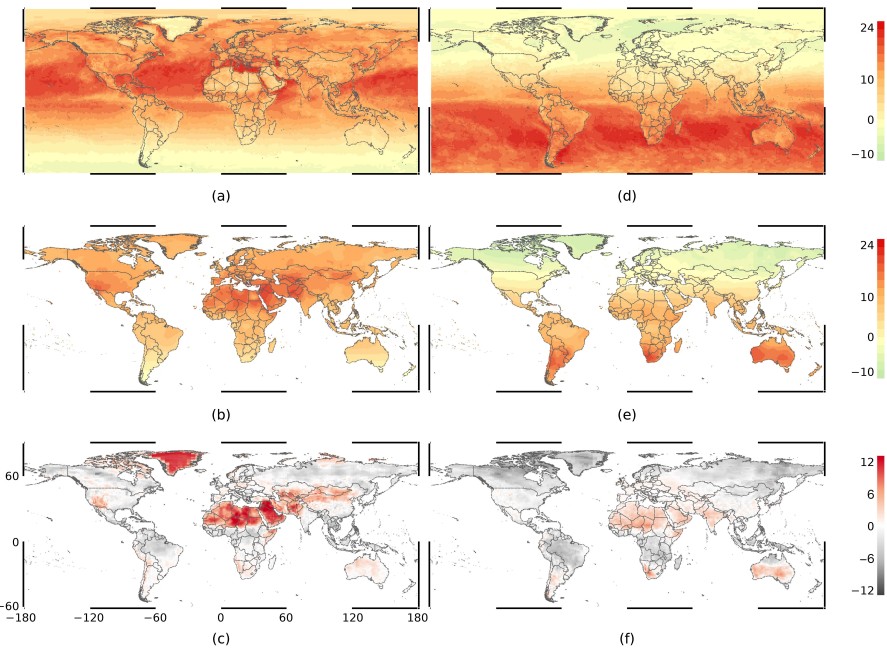

**Figure 6.** Global mean net downward surface radation flux, $\mathrm{MJ\,m^{-2}}$, for June (left) and December (right) 2000 CE from CERES EBAF (top), the SPLASH model (middle), and the differences between SPLASH and CERES EBAF (bottom). (a) CERES EBAF June 2000; (b) SPLASH June 2000; (c) difference between SPLASH and CERES EBAF June 2000; (d) CERES EBAF December 2000; (e) SPLASH December 2000; and (f) difference between SPLASH and CERES EBAF December 2000. Color bars on the right are linear interpolations of results in $\mathrm{MJ\,m^{-2}}$. Differences in red indicate higher SPLASH model results.

for June and December of the same year are plotted in Fig. 7a and 7d, respectively. The relative soil moisture in both datasets is computed as the ratio of $\mathrm{mm}$ of soil moisture over the total bucket size (i.e., 760 $\mathrm{mm}$ in NCEP CPC and 150 $\mathrm{mm}$ in SPLASH).

Overall, the SPLASH model simulates soil moisture patterns similar to the NCEP CPC model results. The differences between the SPLASH and NCEP CPC model results are highlighted in Fig. 7c and 7f. Once again, regions in red indicate where the SPLASH model results are higher than the NCEP CPC model results and regions in gray indicate areas where the

5   SPLASH model results are lower. In contrast to the NCEP CPC soil moisture, the SPLASH model produces a relatively full bucket across wet vegetated regions. The lower relative fullness of the NCEP CPC bucket may be contributed to its significantly larger bucket size. Despite the differing magnitudes of soil moisture, the spatial distributions of soil moisture show consistently drier regions in both simulations at both time periods, especially across mid northern latitudes (e.g., eastern North America, northern Africa, and central Asia). Seasonal shifts in soil moisture from June to December are also consistently shown (e.g.,

10   southern transition in Africa, eastern transition in South America and northern transition in Australia). There are discrepancies in the spatial distribution of soil moisture across the high latitude regions (especially Russia). The predominantly saturated

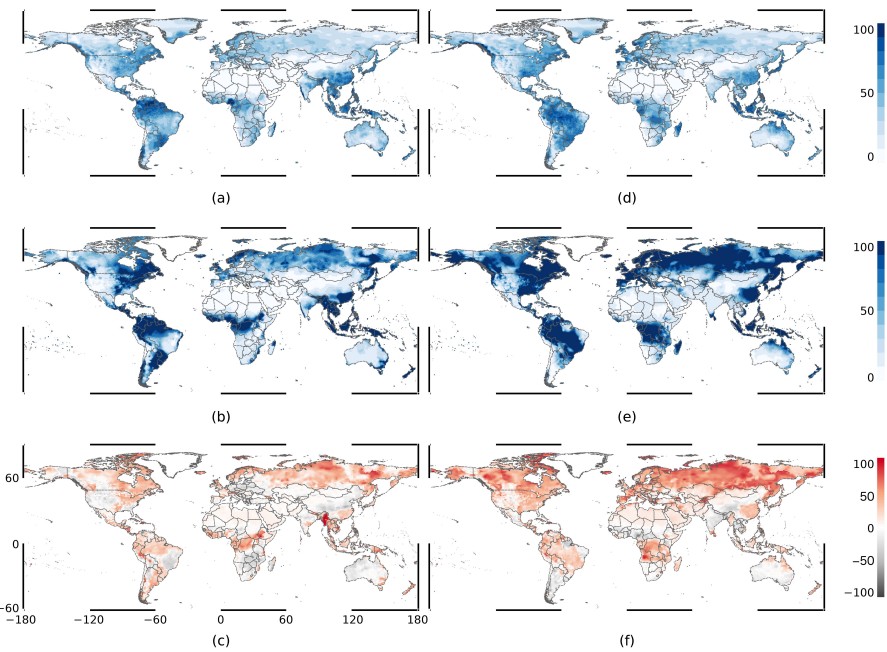

**Figure 7.** Global mean relative soil moisture, %, for June (left) and December (right) 2000 CE from NCEP CPC (top), the SPLASH model (middle), and the difference between SPLASH and NCEP CPC (bottom). (a) NCEP CPC June 2000; (b) SPLASH June 2000; (c) difference between SPLASH and NCEP CPC June 2000; (d) NCEP CPC December 2000; (e) SPLASH December 2000; and (f) difference between SPLASH and NCEP CPC December 2000. Color bars on the right are linear interpolations of results in units of relative soil moisture (%). The relative soil moisture is based on the total bucket size (i.e., 760 mm for NCEP CPC and 150 mm for SPLASH). Differences in red indicate higher SPLASH model results.

conditions in the SPLASH simulations across Russia for December (Fig. 7e) may actually be representative of an increasing snow pack, which could account for these differences.

## 5   Discussion

The results presented in Sect. 4 are intended to illustrate the dynamic patterns and trends in the SPLASH model outputs across regions and seasons for a single year under steady-state. The SPLASH model results are promising despite the model's
5   simplifications and limited climatic drivers. At the local scale, the comparison between SPLASH and VIC across climate and elevation gradients (i.e., Fig. 4) shows relatively good agreement for $E_n^p$. There are some discrepancies between $H_N$, especially at the high-latitude, high-elevation tundra site (i.e., Fig. 4a) and at the low-elevation hot arid desert site (i.e., Fig. 4d), where the SPLASH simulations were higher than VIC for portions to all of the year. These discrepancies are likely due to local deviations from the globally averaged surface albedo. This is especially true when there is snow cover, as SPLASH does not model
10   snowpack. Soil moisture also showed relatively good agreement, except at the equatorial monsoonal site (i.e., Fig. 4e), where

the SPLASH simulation was consistently higher than VIC. This discrepancy may be due to the assumed constant maximum soil moisture holding capacity. Furthermore, at the global scale, the SPLASH model reasonably captures the latitudinal gradation of net surface radiation flux (where surface emission assumptions are valid) compared to the CERES EBAF results (i.e., Fig. 6) and produces similar spatial patterns of soil moisture, albeit at different magnitudes, compared to the NCEP CPC soil moisture results (i.e., Fig. 7).

5      While the methodology presented in Sect. 2 makes numerous assumptions and simplifications (e.g., saturation-excess runoff generation, invariant soil properties, and constant global parameterization), it provides a simple and robust framework for the estimation of radiation components, evapotranspiration, and plant-available moisture requiring only standard meteorological measurements as input. The SPLASH model currently only produces saturation excess runoff. For more realistic runoff gener-ation, other water balance models allow runoff to occur when the bucket is less than full, for example the empirical relationship of runoff to the weighted relative soil moisture in the simple water balance model (Orth et al., 2013). Regarding the bucket size, in principle, $W_m$ in Eq. 21 could be formulated as a property of soil type (as was done, for example, in the original BIOME model), there are some objections to doing so. While $W_m$ has a standard definition in agronomy (i.e., the difference between field capacity and wilting point), the wilting point in reality depends on plant properties. Also, the effective 'bucket size' depends on rooting behavior, which is highly adaptable to the soil wetness profile. The absolute value of daily soil mois-ture will be influenced by the bucket size (as shown in Fig. 7) and can have an impact on the local hydrology (e.g., Fig. 4e); however, plant-available moisture indexes, such as $\alpha$ (i.e., the ratio of supply-limited to non-supply-limited evapotranspiration), have commonly been found to be relatively insensitive to the bucket size. Regarding localized effects, the standard values pre-sented in Table 2 are representative of reasonable global means; however, it is recommended that local parameterization (e.g., shortwave albedo) be used if and when data are available.

20      Over the years, a common misconception has developed regarding the calculation of daily actual evapotranspiration (as defined by Federer, 1982), whereby the integration of Eq. 26 is mistakenly interpreted as:

$$E_n^a = \min\left(S, D\right),\tag{35}$$

where $D$ $(\mathrm{mm\,d^{-1}})$ is the total daily demand, given by Eq. 25, and $S$ $(\mathrm{mm\,d^{-1}})$ is the total daily supply over the hours of positive net radiation, which may be given by:

$$25 \quad S = \int\limits_{day} S_w = \int\limits_{-h_n}^{h_n} S_w = \frac{24}{\pi}\,h_n\,S_w,\tag{36}$$

where $h_n$ is the net radiation cross-over angle, given by Eq. 15, and the constant coefficient converts the units of radians to hours. As shown in Fig. 2, $E_n^a$ is a piecewise function consisting of two curves overlaid throughout the course of a single day that must be accounted for simultaneously; however, even in some recent model developments, $E_n^a$ is calculated using Eq. 35, including the equilibrium terrestrial biosphere models BIOME3 and BIOME4 (Haxeltine and Prentice, 1996; Kaplan, 2001) and the Lund-Potsdam-Jena Dynamic Global Vegetation Model (Sitch et al., 2003). Only under specific circumstances will Eq. 35 produce correct results. It is the intension of this work to provide a simple analytical solution that correctly accounts for the integration of Eq. 26, which has been provided in the form of Eq. 27b.

**Code Availability**

The code, in four programming languages (C++, FORTRAN, Python, and R), is available on an online repository under the GNU Lesser General Public License (https://bitbucket.org/labprentice/splash). The repository includes the present release (v1.0) and working development of the code (with Makefiles where appropriate), example data, and the user manual. All four versions of the code underwent and passed a set of consistency checks to ensure similar results were produced under the same input conditions. The following describes the requirements for compiling and executing SPLASH v.1.0.

For the C++ version, the code was successfully compiled and executed using the GNU C++ compiler (g++ v.4.8.2) pro-vided by the GNU Compiler Collection (Free Software Foundation, Inc., 2016). It utilizes the C numerics library (cmath), input/output operations library (cstdio), and the standard general utilities library (cstdlib) and references the vector container and string type.

For the FORTRAN version, the code was successfully compiled and executed using the PGI Fortran compiler (pgf95 v.16.1-0) provided by The Portland Group - PGI Compilers and Groups (NVIDIA Corporation, 2016) and the GNU Fortran compiler (gfortran v.4.8.4) provided by the GNU Compiler Collection (Free Software Foundation, Inc., 2016).

For the Python version, the code was successfully compiled and executed using Python 2.7 and Python 3.5 interpreters (Python Software Foundation, 2016). It requires the installation of third-party packages, including NumPy (v.1.10.4 by NumPy Developers, 2016) and SciPy (v.0.17.0 by SciPy Developers, 2016) and utilizes the basic date and time types (datetime), logging facility (logging), Unix-style pathname pattern extension (glob), and miscellaneous operating system interfaces (os) modules.

For the R version, the code was successfully compiled and executed using R-3.2.3 "Wooden Christmas-Tree" (The R Foundation for Statistical Computing, 2015).

**Appendix A: Calculating True Longitude**

Berger (1978) presents a method for estimating true longitude, $\lambda$, for a given day of the year, $n$, that associates uniform time (i.e., a mean planetary orbit and constant day of the vernal equinox) to Earth's angular position. The formula is based on classical astronomy and is suitable for calculations in palaeoclimatology. The algorithm begins with the calculation of the mean longitude of the vernal equinox, $\lambda_{m0}$ (rad), assumed to fall on 21 March:

$$\lambda_{m0} = 2\left[\left(\frac{1}{2}\,e + \frac{1}{8}\,e^3\right)(1+\beta)\sin\widetilde{\omega} - \frac{1}{4}\,e^2\left(\frac{1}{2}+\beta\right)\sin 2\widetilde{\omega} + \frac{1}{8}\,e^3\left(\frac{1}{3}+\beta\right)\sin 3\widetilde{\omega}\right], \tag{A1}$$

where $\beta = \sqrt{1-e^2}$. The mean longitude, $\lambda_m$ (rad), is then calculated for a given day based on a daily increment with respect to the day of the vernal equinox (i.e., day 80):

$$\lambda_m = \lambda_{m0} + 2\pi\,(n-80)\,N_a^{-1}, \tag{A2}$$

where $N_a$ is total number of days in the year. The mean anomaly, $\nu_m$ (rad), is calculated based on the equality presented in Eq. 6:

$$\nu_m = \lambda_m - \widetilde{\omega}, \tag{A3}$$

which is then used to determine the true anomaly by:

$$\nu = \nu_m + \left(2e - \frac{1}{4}\,e^3\right)\sin\nu_m + \frac{5}{4}\,e^2\,\sin 2\nu_m + \frac{13}{12}\,e^3\,\sin 3\nu_m, \tag{A4}$$

and is converted back to true longitude by:

$$\lambda = \nu + \widetilde{\omega}. \tag{A5}$$

The resulting $\lambda$ should be constrained to an angle within a single orbit (i.e., $0 \le \lambda \le 2\pi$).

**Appendix B: Calculating Temperature and Pressure Dependencies**

The four variables used to calculate the water-to-energy conversion factor, $E_{con}$, given in Eq. 19 have temperature and/or pressure dependencies that may be solved using the equations presented here.

The temperature-dependent equation for the slope of the saturation vapor pressure-temperature curve, $s$, can be expressed as (Allen et al., 1998):

$$s = \frac{2.503 \times 10^6 \exp\left(\frac{17.27\,T_{air}}{T_{air}+237.3}\right)}{\left(T_{air}+237.3\right)^2}, \tag{B1}$$

where $s$ ranges from about 11 to 393 Pa K$^{-1}$ for $T_{air}$ between $-20$ and $40\,°$C. Please be aware of the typographical error in this formula as presented in Eq. 7 of Gallego-Sala et al. (2010) where 237.3 is misrepresented as 273.3.

The temperature-dependent equation for the latent heat of vaporization, $L_v$, may be expressed as (Henderson-Sellers, 1984):

$$L_v = 1.91846 \times 10^6 \left[\frac{T_{air}+273.15}{(T_{air}+273.15)-33.91}\right]^2, \tag{B2}$$

where $L_v$ ranges from about $2.558 \times 10^6$ to $2.413 \times 10^6$ J K$^{-1}$ for $T_{air}$ between $-20$ and $40\,°$C.

The temperature and pressure dependence of the density of water, $\rho_w$, may be expressed as (Chen et al., 1977):

$$\rho_w = \rho_o \frac{K_o + C_A\,P^*_{atm} + C_B\,{P^*_{atm}}^2}{K_o + C_A\,P^*_{atm} + C_B\,{P^*_{atm}}^2 - P^*_{atm}}, \tag{B3}$$

where $\rho_o$ (kg m$^{-3}$) is the density of water at 1 atm, $K_o$ (bar) is the bulk modulus of water at 1 atm, $C_A$ (unitless) and $C_B$ (bar$^{-1}$) are temperature-dependent coefficients, and $P^*_{atm}$ (bar) is the atmospheric pressure (i.e., $1\,$Pa $= 1 \times 10^{-5}$ bar).

The equation for $\rho_o$ is based on the work of Kell (1975):

$$\rho_o = \sum_{i=0}^{8} C_i\,{T_{air}}^i. \tag{B4}$$

The equation for $K_o$ is also based on the work of Kell (1975):

$$K_o = \sum_{i=0}^{5} C_i\,{T_{air}}^i. \tag{B5}$$

The equations for $C_A$ and $C_B$ are given as (Chen et al., 1977):

$$C_A = \sum_{i=0}^{4} C_i \, T_{air}{}^i, \tag{B6}$$

$$C_B = \sum_{i=0}^{4} C_i \, T_{air}{}^i. \tag{B7}$$

The coefficients for $T_{air}$ in Eqns. B4 through B7 are given in Table 3.

The temperature and pressure dependence of the psychrometric constant, $\gamma$, may be expressed as (Allen et al., 1998):

$$\gamma = \frac{C_p \, M_a \, P_{atm}}{M_v \, L_v}, \tag{B8}$$

where $C_p$ ($\mathrm{J\,kg^{-1}\,K^{-1}}$) is the temperature-dependent specific heat capacity of humid air; $M_a$ ($\mathrm{kg\,mol^{-1}}$) and $M_v$ ($\mathrm{kg\,mol^{-1}}$) are the molecular weights of dry air and water vapor, respectively; $L_v$ ($\mathrm{J\,kg^{-1}}$) is the latent heat of vaporization of water; and $P_{atm}$ (Pa) is the atmospheric pressure. Constants for $M_a$ and $M_v$ are given in Table 2. The temperature dependence of $C_p$ may

be assumed negligible (e.g., $C_p = 1.013 \times 10^3 \, \mathrm{J\,kg^{-1}\,K^{-1}}$) or calculated by (Tsilingiris, 2008):

$$C_p = \sum_{i=0}^{5} C_i \, T_{air}{}^i, \tag{B9}$$

for $T_{air}$ between 0–100 °C. The coefficients of $T_{air}$ are given in Table 3.

*Author contributions.* I. C. Prentice, M. T. Sykes, and W. Cramer developed the original model theory and methods. A. V. Gallego-Sala, B. J. Evans, H. Wang, and T. W. Davis contributed to model improvements. R. T. Thomas, R. J. Whitley, B. D. Stocker, and T. W. Davis

transcribed the new model code and ran simulations. The manuscript was prepared with contributions from all authors.

*Acknowledgements.* This work was primarily funded by Imperial College London as a part of the AXA Chair Programme on Biosphere and Climate Impacts. It is a contribution to the Imperial College initiative on Grand Challenges in Ecosystems and the Environment, and the ecosystem Modelling and Scaling infrasTructure (eMAST) facility of the Australian Terrestrial Ecosystem Research Network (TERN). TERN is supported by the Australian Government through the National Collaborative Research Infrastructure Strategy (NCRIS). BDS funded

by the Swiss National Science Foundation (SNF) and the European Commission's 7th Framework Programme, under Grant Agreement number 282672, EMBRACE project. WC contributes to the Labex OT-Med (n° ANR-11-LABX-0061) funded by the French government through the A∗MIDEX project (n° ANR-11-IDEX-0001-02). AGS has been supported by a Natural Environment Research Council grant (NERC grant number NE/I012915/1). VIC simulations utilized the Janus supercomputer, which is supported by the National Science Foundation (award number CNS-0821794) and the University of Colorado Boulder. The Janus supercomputer is a joint effort of the University of Col-

orado Boulder, the University of Colorado Denver and the National Center for Atmospheric Research. CERES EBAF data were obtained from the NASA Langley Research Center Atmospheric Science Data Center. CPC Soil Moisture data provided by the NOAA/OAR/ESRL PSD, Boulder, Colorado, USA, from their website at http://www.esrl.noaa.gov/psd/.

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

**Table 1.** Nomenclature.

| | Instantaneous |
|---|---|
| $S_w$ | evaporative supply rate, $\mathrm{mm\,h^{-1}}$ |
| $D_p$ | evaporative demand rate, $\mathrm{mm\,h^{-1}}$ |
| $E^q$ | equilibrium evapotranspiration rate, $\mathrm{mm\,h^{-1}}$ |
| $E^p$ | potential evapotranspiration rate, $\mathrm{mm\,h^{-1}}$ |
| $E^a$ | actual evapotranspiration rate, $\mathrm{mm\,h^{-1}}$ |
| $I_o$ | extraterrestrial solar radiation flux, $\mathrm{W\,m^{-2}}$ |
| $I_N$ | net radiation flux, $\mathrm{W\,m^{-2}}$ |
| $I_{SW}$ | net shortwave solar radiation flux, $\mathrm{W\,m^{-2}}$ |
| $I_{LW}$ | net longwave radiation flux, $\mathrm{W\,m^{-2}}$ |

| | Daily |
|---|---|
| $W_n$ | soil moisture, mm |
| $P_n$ | precipitation, $\mathrm{mm\,d^{-1}}$ |
| $C_n$ | condensation, $\mathrm{mm\,d^{-1}}$ |
| $RO$ | runoff, mm |
| $E_n^q$ | equilibrium evapotranspiration, $\mathrm{mm\,d^{-1}}$ |
| $E_n^p$ | potential evapotranspiration, $\mathrm{mm\,d^{-1}}$ |
| $E_n^a$ | actual evapotranspiration, $\mathrm{mm\,d^{-1}}$ |
| $H_o$ | solar irradiation, $\mathrm{J\,m^{-2}\,d^{-1}}$ |
| $H_N$ | net surface radiation, $\mathrm{J\,m^{-2}\,d^{-1}}$ |
| $H_N^+$ | positive net surface radiation, $\mathrm{J\,m^{-2}\,d^{-1}}$ |
| $H_N^-$ | negative net surface radiation, $\mathrm{J\,m^{-2}\,d^{-1}}$ |
| $Q_n$ | photosynthetically active radiation, $\mathrm{mol\,m^{-2}\,d^{-1}}$ |
| $S_f$ | fraction of bright sunshine hours, unitless |
| $T_{air}$ | mean air temperature, °C |

| | Monthly |
|---|---|
| $E_m^q$ | equilibrium evapotranspiration, $\mathrm{mm\,mo^{-1}}$ |
| $E_m^p$ | potential evapotranspiration, $\mathrm{mm\,mo^{-1}}$ |
| $E_m^a$ | actual evapotranspiration, $\mathrm{mm\,mo^{-1}}$ |
| $\Delta E_m$ | climatic water deficit, $\mathrm{mm\,mo^{-1}}$ |
| $\alpha_m$ | Priestley-Taylor coefficient, unitless |

**Table 1 (continued).** Nomenclature.

| | Miscellaneous |
|---|---|
| $\cos\theta_z$ | inclination factor, unitless |
| $\delta$ | declination angle, rad |
| $d_r$ | distance factor, unitless |
| $\varepsilon$ | obliquity, rad |
| $e$ | eccentricity, unitless |
| $E_{con}$ | water to energy conversion factor, $\mathrm{m^3\,J^{-1}}$ |
| $\gamma$ | psychrometric constant, $\mathrm{Pa\,K^{-1}}$ |
| $h$ | hour angle, rad |
| $h_i$ | intersection of evaporative rates hour angle, rad |
| $h_n$ | net radiation crossover hour angle, rad |
| $h_s$ | sunset hour angle, rad |
| $i$ | day of month (1–31) |
| $\lambda$ | true longitude, rad |
| $L_v$ | latent heat of vaporization of water, $\mathrm{J\,kg^{-1}}$ |
| $\nu$ | true anomaly, rad |
| $n$ | day of year (i.e., 1–365) |
| $N_a$ | total number of days in a year (e.g., 365) |
| $N_m$ | total number of days in a given month (e.g., 31) |
| $\widetilde{\omega}$ | longitude of perihelion, rad |
| $\phi$ | latitude, rad |
| $P_{atm}$ | atmospheric pressure, Pa |
| $\rho_w$ | density of water, $\mathrm{kg\,m^{-3}}$ |
| $r_u$ | $\sin\delta\,\sin\phi$, unitless |
| $r_v$ | $\cos\delta\,\cos\phi$, unitless |
| $r_w$ | $(1-\beta_{sw})\,\tau\,I_{sc}\,d_r$, $\mathrm{W\,m^{-2}}$ |
| $r_x$ | $3.6\times10^6\,(1+\omega)\,E_{con}$, $\mathrm{mm\,m^2\,W^{-1}\,h^{-1}}$ |
| $s$ | slope of saturated vapor pressure-temperature curve, $\mathrm{Pa\,K^{-1}}$ |
| $\tau$ | transmittivity, unitless |
| $\tau_o$ | transmittivity at mean sea level, unitless |
| $z$ | elevation above mean sea level, m |

**Table 2.** Constants and Standard Values.

| Variable | Units | Description |
|---|---|---|
| $A$ | 107 °C | empirical constant, Eq. 13 (Monteith and Unsworth, 1990) |
| $\beta_{sw}$ | 0.17 | shortwave albedo, Eq. 10 (Federer, 1968) |
| $\beta_{vis}$ | 0.03 | visible light albedo, Eq. 17 (Sellers, 1985) |
| $b$ | 0.20 | empirical constant, Eq. 13 (Linacre, 1968) |
| $c$ | 0.25 | cloudy transmittivity, Eq. 12 (Linacre, 1968) |
| $d$ | 0.50 | angular coefficient of transmittivity, Eq. 12 (Linacre, 1968) |
| fFEC | 2.04 µmol J$^{-1}$ | flux-to-energy conversion, Eq. 17 (Meek et al., 1984) |
| $g$ | 9.80665 m s$^{-2}$ | standard gravity, Eq. 20 (Allen, 1973) |
| $I_{sc}$ | 1360.8 W m$^{-2}$ | solar constant, Eq. 2 (Kopp and Lean, 2011) |
| $L$ | 0.0065 K m$^{-1}$ | mean adiabatic lapse rate, Eq. 20 (Allen, 1973) |
| $M_a$ | 0.028963 kg mol$^{-1}$ | molecular weight of dry air, Eq. 20 (Tsilingiris, 2008) |
| $M_v$ | 0.01802 kg mol$^{-1}$ | molecular weight of water vapor, Eq. B8 (Tsilingiris, 2008) |
| $\omega$ | 0.26 | entrainment factor, Eq. 22 (Priestley and Taylor, 1972) |
| $P_o$ | 101325 Pa | standard sea-level pressure, Eq. 20 (Allen, 1973) |
| $R$ | 8.31447 J mol$^{-1}$ K$^{-1}$ | universal gas constant, Eq. 20 (Moldover et al., 1988) |
| $S_c$ | 1.05 mm h$^{-1}$ | supply rate constant, Eq. 21 (Federer, 1982) |
| $T_o$ | 288.15 K | base temperature, Eq. 20 (Berberan-Santos et al., 1997) |
| $W_m$ | 150 mm | soil moisture capacity, Eq. 21 (Cramer and Prentice, 1988) |

**Table 3.** Coefficients of $T_{air}$.

| | $\rho_o$ (kg m$^{-3}$) Eq. B4 | $K_o$ (bar) Eq. B5 | $C_A$ (unitless) Eq. B6 | $C_B$ (bar$^{-1}$) Eq. B7 | $C_p$ (J kg$^{-1}$ K$^{-1}$) Eq. B9 |
|---|---|---|---|---|---|
| $C_0$ | $+9.998395 \times 10^2$ | $+1.96520 \times 10^4$ | $+3.26138$ | $+7.2061 \times 10^{-5}$ | $+1.004571 \times 10^3$ |
| $C_1$ | $+6.78826 \times 10^{-2}$ | $+1.48183 \times 10^2$ | $+5.223 \times 10^{-4}$ | $-5.8948 \times 10^{-6}$ | $+2.050633$ |
| $C_2$ | $-9.08659 \times 10^{-3}$ | $-2.29995$ | $+1.324 \times 10^{-4}$ | $+8.6990 \times 10^{-8}$ | $-1.631537 \times 10^{-1}$ |
| $C_3$ | $+1.02213 \times 10^{-4}$ | $+1.28100 \times 10^{-2}$ | $-7.655 \times 10^{-7}$ | $-1.0100 \times 10^{-9}$ | $+6.212300 \times 10^{-3}$ |
| $C_4$ | $-1.35439 \times 10^{-6}$ | $-4.91564 \times 10^{-5}$ | $+8.584 \times 10^{-10}$ | $+4.3220 \times 10^{-12}$ | $-8.830479 \times 10^{-5}$ |
| $C_5$ | $+1.47115 \times 10^{-8}$ | $+1.03553 \times 10^{-7}$ | — | — | $+5.071307 \times 10^{-7}$ |
| $C_6$ | $-1.11663 \times 10^{-10}$ | — | — | — | — |
| $C_7$ | $+5.04407 \times 10^{-13}$ | — | — | — | — |
| $C_8$ | $-1.00659 \times 10^{-15}$ | — | — | — | — |