# Peer review of "Simple Process-Led Algorithms for Simulating Habitats (SPLASH v.1.0): Robust Indices of Radiation, Evapotranspiration and Plant-Available Moisture"

_Geoscientific Model Development, 2016_

## Referee Comment (RC1) · Anonymous Referee #1 · 10 May 2016

This paper describes a process-based model for some surface flux and other quantity reconstructions, such as radiation, evapotranspiration, and soil moisture, using three daily mean meteorological inputs (i.e., near-surface air temperature, precipitation, and either fraction of bright sunshine hours or fractional cloud cover), latitude, and elevation, in order to overcome data deficiency of plant ecophysiological and biogeophysical studies. The process-based model that is based on theoretical understanding of relevant environmental processes can provide a practical opportunity for understanding specific responses to changes in environmental conditions. For example, the model can be used to estimate the response of paleoenvironmental indicators to the really

different climates from the present a situation that statistical models might make inappropriate extrapolations. Therefore, the manuscript will be really useful for the community to implement paleoclimate and paleoecological analysis.

I realize that this is a consolidated study of separately developed modules in the STASH model, and not a really "new model" paper. Overall, this manuscript is well written and clearly describes the calculation processes. But, the process-based model is greatly simplified; therefore you need to validate whether the model output is reliable or not using the globally or locally observed data. Although I do not know any observed evapotransipiration and plant-available moisture data, we have some monthly-observed radiation data at the surface and top-of-atmosphere (e.g. Clouds and Earth's Radiant Energy System (CERES)). Using the data set, you could validate what extent the simplified process-based model simulates radiation terms.

In the result section (P. 13, L11- ), you run the model using a grid data above San Francisco. If you access monthly-observed radiation, evapotranspiration, and soil moisture data there, you can keep the results and then you should add the observed ones into Figure 4. Then, you can discuss the model performance. If not, you should choose other locations where you can access the observed data.

With regard to inputs, do we really need mean daily meteorological variables for the robust approximations of key quantities? If you really need the mean daily data, you should choose any reanalysis data (e.g., ERA-interim daily time data) for the all three meteorological data. I do not understand why you use the mixture of daily and monthly data well. Then, long-term monthly mean values (in equilibrium past climate states, such as mid-Holocene and Last Glacial Maximum) are common for paleoclimate community. Therefore, if you expect the application of the model for paleoclimate studies, you should make a conversion module (i.e., monthly data into quasi-daily data). It is not difficult for you to put the module into the SPLASH code because this kind of model is included BIOME3, BIOME4, and LPJ DGVM. Finally, what do you think if we use quasi-daily data for running the model, it estimates really different output from the one

with regular daily time-step meteorological data?

Other minor comments:

P. 1, L5, what are the exact time scales of "ecologically relevant time scales"? Under your assumption, can we use a traditional climatology (i.e., 30-year average) data?

P. 3, L6, change "air temperature" to "near-surface air temperature" or "air temperature at the height of 2m"

P. 3, L27, one term/character missing for "the analytical integral of the minimum ... over a single day"

P. 5, L25, surface shortwave albedo is constant (0.17) for the model simplification, but is it okay for the energy balance at local? The surface condition including albedo must be different at local. Therefore, the anomaly (actual surface albedo at local minus 0.17) largely impacts on surface net radiation and thus evapotranspiration and soil moisture in some regions, right?

P. 13, L13, although the model requires daily meteorological input (P. 3, L6-7), why do you use monthly cloud fraction data here? Do you think that cloud fraction is less significant for the calculation, compared to the other meteorological variables? I think that the diversity of the forms of clouds and their strong spatial and temporal variability determine the dynamics of the radiation budget to a significant degree.

―――――――――――――――――――

---

## Referee Comment (RC2) · Anonymous Referee #2 · 25 May 2016

This manuscript presents the new SPLASH model, which builds on the widely-used STASH model for calculating bioclimatic variables. The authors describe improvements to the model algorithms that resolve some known issues with STASH. In addition, the authors have also made available the model code in C++, FORTRAN, Python, and R versions. Both the manuscript and model code will be very useful to the climate and ecological modelling communities. Below I have listed some general comments followed by more specific comments. For this review, I ran the FORTRAN version of SPLASH using the San Francisco test data described in the manuscript. I also ran global 10-minute simulations using 30-year mean pseudo-daily climate data. The FOR-

TRAN code is well documented and was easy to compile (I used an INTEL FORTRAN compiler) and run.

As currently written, the manuscript presents model results for one grid point near San Francisco, California (USA). I would suggest expanding the manuscript to include figures displaying model output for the globe (these could be added to the text or placed in a supplement) as well as some evaluation of the results. As noted in the text, this model can (and will) be applied to spatial grids so it is important to provide evidence that the model works across the range of global climate conditions. Evaluating the model output using observed data is also important. SPLASH uses a set of simplified equations and so the model output will not necessarily closely match observed data. Instead the importance of the evaluation is that it will provide some indication of the extent to which SPLASH may over- or under-predict certain variables and whether there are any spatial biases in the model performance.

As you note (page 2, lines 31-34), SPLASH has been designed so that it can be used for palaeoclimate applications. The code is currently set up so that the orbital parameters used for palaeoclimate simulations are not input as variables but are specified in the code as parameters. I would suggest adding text to the manuscript (and a comment in the code) that describes for the user exactly what variables need to be modified when running the model for palaeo time periods (e.g., obliquity, eccentricity, longitude of perihelion). It would also be useful to describe for the reader any modifications that are required to the input data when doing palaeoclimate simulations. For example, for a palaeoclimate simulation, how should a user specify the input year so that it works with the Julian day subroutine (i.e., get_julian_day)? If a user was running a simulation for 4 ka (∼2000 BCE) would they need to specify the input year as -2000 to get the correct Julian day? Does the Meeus (1991) Julian day algorithm used in the model work for negative numbers?

Page 1, line 11: You say that the climatic drivers for the model include "either fraction of bright sunshine hours or fractional cloud cover," which makes the two variables sound

interchangeable as input. The equations in the code are set up to use sunshine data as input, not cloud cover. I would suggest deleting the reference in this sentence to fractional cloud cover to prevent a user from misinterpreting this sentence as indicating that cloud cover data can be used directly without having to first convert them to sunshine data.

Page 3, lines 26-27, Equation 1: As described in Section 2.6, runoff is subtracted from the water balance in calculating daily soil moisture so I would suggest including runoff as a term in Equation 1 (i.e., subtracting RO in the right hand side of the equation).

Page 8, lines 24-30: You list some objections to using observed soil properties to estimate maximum soil moisture capacity, including that doing so does not change the seasonal course of soil moisture. I do not think there is any problem with specifying a constant size (in this case, 150 mm) for the soil moisture "bucket" as for some research questions it could be important to hold bucket size constant. However, the seasonal pattern of soil moisture does change depending on the bucket size. I ran SPLASH for the San Francisco grid point using a 100 mm bucket and the soil moisture reached saturation sooner in the year and also was depleted earlier in the spring. Similarly, using a 200 mm bucket, the soil moisture reached saturation later in the year and more soil moisture was available later in the growing season. These seasonal shifts in soil moisture availability may be significant because, for example, they can affect calculations of annual water deficit. I would suggest revising the text stating that the course of seasonal soil moisture is insensitive to the bucket size, as well as the text indicating that soil type-dependent values for the bucket would not improve the model accuracy.

Page 13, line 13: Add the name of the specific WATCH data set you used.

Page 17, line 10: I would add text describing the typographical error (e.g., Eq. 7 of Gallego-Sala et al. (2010) used 273.3 instead of 237.3).

Code: The SPLASH code is currently available for download from an online Git repository. I would suggest providing the code in a supplemental file accompanying the manuscript so that the code is documented in case the Git repository is unavailable in the future. The user could still be directed to the online Git repository for the most current version of the code.

The following comments refer to the FORTRAN version of the code, although the same issues may be present in the C++, R, and Python versions.

1. There are a number of debugging comments in the code I downloaded from the Git repository, such as "consistency check – XXX PROBLEM: THIS LEADS TO DIF-FERENCE WITH OTHER VERSIONS XXX." To prevent confusion for the user, remove these comments if the issues have been resolved. If the issues have not been resolved, provide enough detail in the code comment so that the user can determine how the issue may affect their results.

2. In various places in the code the user is referred to particular equations in the documentation file (splash_doc.pdf) that accompanies the code. However, in some cases the equation referenced in the code does not match the equation in the documentation. For example, the calculation of daily photosynthetic photon flux density (PPFD) in the FORTRAN code refers the reader to Equation 57 in the documentation file, which is an equation for the bulk modulus of water. Check that the references in the code to the splash_doc.pdf file are correct.

3. It would help the user if all of the variable names were defined in the code. For example, in the code where PPFD is defined as a type real variable the accompanying comment defines PPFD as "daily PPFD (mol/mˆ2)" instead of "daily photosynthetic photon flux density (mol/mˆ2)."

Figure 3: Change "CRU TS" to "CRU TS3.21" in the caption text.

[Figure]

---

## Author Comment (AC1) · 7 Sep 2016

**Response to Reviewers**

**Title:** Simple Process-Led Algorithms for Simulating Habitats (SPLASH v.1.0): Robust Indices of Radiation, Evapotranspiration and Plant-Available Moisture

**Authors:** T. W. Davis, I. C. Prentice, B. D. Stocker, R. T. Thomas, R. J. Whitley, H. Wang, B. J. Evans, A. V. Gallego-Sala, M. T. Sykes, and W. Cramer

**Manuscript Number:** GMD-2016-49

We thank the reviewers for their constructive and valuable comments. Below, we have responded to each of the comments and concerns. For clarity, we have included a summary of the comments in italics. We would also like to point out the following additional changes to the manuscript:

1. We have added a new co-author, Rebecca T. Thomas, who is responsible for coding and producing the global SPLASH simulations.
2. We moved the variable substitutions, $r_u$ and $r_v$, from Eq. 8 into Eq. 7 to improve the clarity of the later equation derivations.
3. We have updated the definition of the net surface radiation, $H_N$, in Section 2.1.2 for clarity, such that it now encapsulates a full 24-hour period (not just daytime), and have separated it into its positive, $H_N+$, and negative, $H_N-$, components. We have updated the notation of negative net radiation, from $H_N*$ to $H_N-$, in Eqns. 16 and 18. The distinction between $H_N$ and $H_N+$ is also made in Eq. 24 in Section 2.4 as well as in Table 1. We have also made corrections to Eq. 16 for the negative net radiation.
4. Eq. 35 in the Discussions has been moved to Section 2, as it seemed more appropriate to make note of it here.
5. Figures 1 and 2 have been updated for improved clarity.
6. Figure 4 has been updated to match the style of the new Figure 3.
7. The results section has been broken into two subsections: one for local and one for global simulations.

Reviewer #1

**Comment #1.1:**

> *"[...] the process-based model is greatly simplified; therefore you need to validate whether the model output is reliable or not using globally or locally observed data."*
>
> *The reviewer suggests that despite the lack of observations of evapotranspiration and plant-available moisture, there are observations of surface and top-of-the-atmosphere radiation available (e.g., CERES) that could be compared to the SPLASH model radiation simulations. The reviewer notes that observations of monthly radiation, evapotranspiration, and soil moisture should accompany Fig. 4, else another site should be selected where data is available.*

In our search for daily hydro-meteorological data, we discovered the publication by Ben Livneh and others (doi: 10.1038/sdata.2015.42) consisting of 1/16th degree gridded data over North America (i.e., Southern Canada, United States, and Mexico) that includes observations (e.g., precipitation, maximum and minimum air temperature) from NOAA National Climate Data Center's (NCDC) Global Historical Climatology Network (GHCN) along with simulated fluxes (e.g., net radiation, runoff, evapotranspiration, and soil moisture) produced by the VIC model. We have plotted these data along side our own for comparison of local scale trends (see figure below). The figure and analysis have been updated in the manuscript.

[Figure]

The monthly sunshine fraction (Fig 3a) is unchanged. Fig. 3b now shows the SPLASH net radiation flux along with the VIC model net radiation from Livneh et al. (2015), converted from units of W m$^{-2}$ to MJ m$^{-2}$ (red line). The SPLASH net radiation exhibits slightly higher values, particularly in the later season, likely due to the skewness of the monthly sunshine fraction. Fig. 3c results have changed slightly due to a correction in the negative net radiation equation. Fig. 3d now includes daily precipitation from Livneh et al. (red line), which has some higher peak values during the winter months, but otherwise is relatively consistent in the timing of rain events to the daily WATCH precipitation (black line) throughout the year. Fig. 3e shows results from Livneh et al. (2015) for the three-layer VIC model (red lines) and SPLASH (black line). SPLASH consistently models soil moisture between the first and second layers (red solid and dashed lines, respectively), except for the rainy season where the SPLASH soil moisture is higher in magnitude, now between the second and third layers (red dashed and dotted lines, respectively). Fig. 3f indicates that SPLASH runoff (black line) is higher in magnitude at peak rainfall events during the wet season compared to the VIC modeled runoff (in red). In Fig. 3g, the red colored region depicts the range of minimum to maximum near-surface air temperature from Livneh et al., which does fairly well at enveloping the daily WATCH mean near-surface air temperature (black line). Lastly, in Fig. 3h, the VIC potential evapotranspiration curve (red line) shows a consistent seasonal course compared to the SPLASH potential evapotranspiration curve (solid black line), albeit with a significantly higher variability during the summer months.

We took the advice of the reviewer and acquired global CERES net surface radiative fluxes, which we have plotted beside the SPLASH model simulations for the months of June and December 2000 (see figure below).

[Figure]

The top row shows the monthly mean net downward surface radiative flux, MJ m$^{-2}$, for June 2000 and the bottom row for December. The left column are the results from the CERES EBAF and the right column are from SPLASH. As shown in the figure, SPLASH does a reasonably good job at capturing the latitudinal gradients and temporal shifts of net surface radiation. The hot-spots that SPLASH simulates over the deserts and tundra, which are not seen in the CERES EBAF results, are likely do to the invalidation of the well-watered constant surface emission assumption.

**Comment #1.2:**

> *"[...] do we really need mean daily meteorological variables for the robust approximations of key quantities?"*
>
> *The reviewer questions the validity of mixing monthly and daily inputs as is done in the results section (i.e., monthly cloudiness is used in combination with daily precipitation and air temperature). The reviewer notes that for paleoclimate studies, it is common to have mean monthly data and, therefore, recommends implementing a quasi-daily conversion of monthly data to meet the daily input requirements of SPLASH. The necessity of implementing quasi-daily conversion of monthly inputs for paleoclimate studies raises the question of the influence that the quasi-daily conversion will have on the model outputs when compared to regular daily-time stepped meteorological data.*

We thank the reviewer for mentioning this, as we did not clarify our choice of input datasets in the manuscript. It is correct that, in most cases, mean monthly meteorology will be the only datasets available for driving the model, and to answer the question, mean daily data is *not* required. Traditionally, as it was noted, quasi-daily methods are used to convert mean monthly quantities to proximal daily values. This is still a viable and recommended method for producing the input datasets. However, for explanatory purposes, we chose to use daily WATCH meteorology for precipitation and near-surface air temperature, as they were available to us and also to help illustrate how daily variability propagates through the model.

Further to the point in regards to quasi-daily methods, the choice of methodology can/will affect model results. As there are varied methods for producing quasi-daily data (e.g., constant, linear interpolation, empirical model, weather generator), we leave this decision and its influence on the model results to the user. We have included text in Section 1 and Section 4 of the manuscript to explain our choice of datasets and to mention the use of quasi-daily methods in lieu of daily meteorological observations. For temperature and cloudiness, we feel that the constant daily or linearly interpolated values will produce reasonably similar results; however, to capture the stochastic nature of precipitation, a weather generator may be required,

which may be included in later versions of the SPLASH model.

**Comment #1.3**:

> *"P. 1, L5, what are the exact time scales of 'ecologically relevant time scales'? Under your assumption, can we use a traditional climatology (i.e., 30-year average) data?"*

By "ecologically relevant" we mean time scales from months to decades. SPLASH can be driven with transient (month-by-month) data, if used in the modeling of e.g. carbon and water fluxes or tree rings. It can equally be driven with a multi-decadal average climatological seasonal cycle, if used in the modeling of geographic distributions of functional traits or species. We have added these examples of ecological time scales in the text of Section 1.

**Comment #1.4:**

> *"P. 3, L6, change 'air temperature' to 'near-surface air temperature' or 'air temperature at the height of 2m'"*

We thank the reviewer for noting this correction; it has been addressed in the text of Section 1. This distinction has also been noted in the abstract.

**Comment #1.5:**

> *"P. 3, L27, one term/character missing for 'the analytical integral of the minimum ... over a single day'"*

We thank the reviewer for noting this. This was a typesetting mistake and has been corrected.

**Comment #1.6**:

> *"P. 5, L25, surface shortwave albedo is constant (0.17) for the model simplification, but is it okay for the energy balance at the local scale? The surface condition including albedo must be different locally. Therefore, the anomaly (actual surface albedo at local minus 0.17) largely impacts on surface net radiation and thus evapotranspiration and soil moisture in some regions, right?"*

This is a good point. Changes in the shortwave albedo will influence the net radiation, which in turn influences actual evapotranspiration and condensation and, therefore, influences soil moisture. A quick sensitivity analysis of shortwave albedo showed only subtle changes in these variables when the value was halved (i.e., 0.08); however, this was not comprehensively analyzed in this study.

The purpose of the global constants is for model simplification. Their values represent reasonable "global means" and provide an approximation should the researcher have no other information to go on. However, these values can be specified for a locality, if and when the data are available. Due to the rarity of global datasets of such values, we feel assigning constant values for the sake of simplicity is justifiable. We have added text in Section 5 that clarifies the use of localized datasets in replacement of global constants.

**Comment #1.7:**

> *"P. 13, L13, although the model requires daily meteorological input (P. 3, L6-7), why do you use monthly cloud fraction data here? Do you think that cloud fraction is less significant for the calculation, compared to the other meteorological variables? I think that the diversity of the forms of clouds and their strong spatial and temporal variability determine the dynamics of the radiation budget to a significant degree."*

As mentioned in the response to Comment #1.2, we chose mean daily meteorology because of its availability and to help exemplify model results. The use of monthly cloudiness was partly out of convenience and partly due to the absence of a better alternative. We did not use monthly cloudiness because we thought it less significant.

Reviewer #2

**Comment #2.1:**

> The reviewer suggests *"expanding the manuscript to include figures displaying model output for the globe [...] as well as some evaluation of the results [... as ...] it is important to provide evidence that the model works across the range of global climate conditions."*
>
> The reviewer further notes the importance of evaluating model performance such that other researchers have an indication as to whether SPLASH over- or under-estimates certain variables or has any spatial biases.

We have investigated the literature for global observations for comparing against the SPLASH model. The first comparison is with CERES EBAF surface net downward radiative flux (described in Reviewer Comment #1.1). The second comparison is with NCEP CPC V2 soil moisture (van den Dool et al., 2003; doi: 10.1029/2002JD003114) and is shown in the following figure.

[Figure]

The top row shows the relative mean daily soil moisture, %, for June 2000 and the bottom row for December. The left column is the NCEP CPC soil moisture and right column is from SPLASH (following a spin-up to equilibrate the soil moisture fields). We have plotted the relative soil moisture instead of the magnitude due to the significant differences in bucket size (i.e., 760 mm in NCEP CPC and 150 mm in SPLASH). The SPLASH soil moisture simulations result in a relatively full bucket throughout the wet regions. We contribute the comparatively empty bucket in the NCEP CPC soil moisture results to its larger bucket size. Nevertheless, the spatial patterns and seasonal shifts of soil moisture is consistent between the two models. There is a bias in the SPLASH soil moisture in the north-eastern region of Russia, which may be due to the lack of a long-term spin-up of soil moisture.

**Comment #2.2:**

> *"[...] SPLASH has been designed so that it can be used for palaeoclimate applications [... however, the ...] code is currently set up so that the orbital parameters used for palaeoclimate simulations are not input as variables but are specified [...] as parameters."*
>
> *The reviewer suggests that these palaeoclimate-specific parameters (i.e., obliquity, eccentricity and longitude of the perihelion) be clarified in the manuscript and that a description be added as to how these parameters may be changed in the code for palaeo applications. The reviewer further requests that an explanation be presented on how users are meant to input dates for palaeoclimate studies (e.g., does the Meeus (1991) Julian day algorithm work for negative years?)*

Thank you for making this clear to us. The code has been updated to identify the paleoclimate variables; however, it is up the user as to which is the appropriate method for updating these values—we made reference to Berger (1978) and Berger & Loutre (1991) for possible algorithms in Section 2.1.1. We added a note on our use of Berger (1978) to calculate the constant orbital parameters in our results.

In Meeus (1991), the Julian day algorithm is valid for positive and negative Gregorian calendar years, but not for negative Julian days (i.e., invalid for dates on or before noon −4712 January 1). Other methods exist, such as ignoring leap years. We understand the difficulty of tracking individual dates into the far past and admit that the current implementation in SPLASH only partially addresses the needs of paleoclimatological studies. The limitation of the Meeus algorithm has been explicitly defined in the source code.

**Comment #2.3:**

> *The reviewer requests clarity be added to the required climatic drivers in regards to the use of fractional bright sunshine hours or fractional cloud cover, which currently reads as interchangeable quantities— actual model input requires fractional bright sunshine hours.*

The ambiguity regarding the required climatic drivers has been addressed in the text of the abstract. We feel that in the introduction and results, our explanation of the difference between fractional bright sunshine hours and cloudiness fraction is sufficient for readers to understand.

**Comment #2.4:**

> *The reviewer notes that Eq. 1 is missing the runoff term (as described in §2.6).*

Originally, we had lumped runoff with the correction of daily soil moisture. However, it is clearer to the reader, especially in regards to the bucket model, if we include the runoff parameter in Eq. 1, instead of as a by-product of our soil moisture calculation. Therefore, we have corrected Eq. 1 in Section 2, such that it now includes runoff as a parameter. To accommodate this, we have added a new subsection for runoff in the methodology (Section 2.6) and have amended the text in Section 2.7.

**Comment #2.5:**

> *The reviewer points out that one of the objections made against a soil-dependent bucket size (i.e., that the seasonal course of soil moisture is insensitive to the exact value specified) may not be sufficient, as the seasonal pattern of soil moisture does, in fact, change with respect to a changing bucket size, which could have implications on the annual water deficit (and/or phenology).*

The absolute values of modeled soil moisture are influenced by the bucket size, which may also influence the soil moisture memory (this may be important in long-term simulations). However, the evapotranspiration (ET) rate is determined by the

atmospheric demand and the *fractional* volumetric soil moisture content. In many applications, the quantity α (the ratio of supply-limited to non-supply-limited ET) is used as an index of water availability, and it has commonly been found that this value is insensitive to the bucket size. We have amended our defense of using a soil-independent bucket size and have moved it from Section 2.3 to the Discussions.

**Comment #2.6:**

> *The reviewer suggests uploading the release version of the code (in all of its forms) to the journal in addition to hosting it on the Git repository to make certain of its future preservation.*

Thank you for this comment. We plan to upload our source code to GMD for accessibility and preservation.

**Comment #2.7:**

> *Page 13, line 13: Add the name of the specific WATCH data set you used.*

Thank you for this comment; we have addressed this ambiguity by noting the use of the WATCH Forcing Data methodology applied to the ERA-Interim, first release, 2012.

**Comment #2.8:**

> *Page 17, line 10: I would add text describing the typographical error (e.g., Eq. 7 of Gallego-Sala et al. (2010) used 273.3 instead of 237.3).*

Thank you for this comment; as per your recommendation, the typographical error has been noted.

**Comment #2.9:**

> *Figure 3: Change "CRU TS" to "CRU TS3.21" in the caption text.*

Thank you for this suggestion. Both the CRU TS and WFDEI data source versions have been added to the figure caption to improve their clarity.

**Comment #2.10:**

> *There are a number of debugging comments in the code I downloaded from the Git repository, such as "consistency check – XXX PROBLEM: THIS LEADS TO DIFFERENCE WITH OTHER VERSIONS XXX." To prevent confusion for the user, remove these comments if the issues have been resolved. If the issues have not been resolved, provide enough detail in the code comment so that the user can determine how the issue may affect their results.*

Our apologies; the old debugging comments have been removed from the source code.

**Comment #2.11:**

> *In various places in the code the user is referred to particular equations in the documentation file (splash_doc.pdf) that accompanies the code. However, in some cases the equation referenced in the code does not match the equation in the documentation. For example, the calculation of daily photosynthetic photon flux density (PPFD) in the FORTRAN code refers the reader to Equation 57 in the documentation file, which is an equation for the bulk modulus of water. Check that the references in the code to the*

*splash_doc.pdf file are correct.*

Thank you for noting this. Outdated references to the documentation have been removed from the source code.

**Comment #2.12:**

*It would help the user if all of the variable names were defined in the code. For example, in the code where PPFD is defined as a type real variable the accompanying comment defines PPFD as "daily PPFD (mol/mˆ2)" instead of "daily photosynthetic photon flux density (mol/mˆ2)."*

Thank you for this suggestion. Variable names throughout the code have been written out in the comments to help identify them.

---

## Author Comment (AC2) · 27 Sep 2016

**Addendum**

In our Consolidated Response to referee comments, we noted a discrepancy between the simulated patterns of soil moisture in northeastern Russia and the patterns shown in the NCEP re-analysis. The SPLASH-simulated soil moisture Figure presented there contained unrealistically sharp boundaries which we had previously attributed to a spin-up issue. However, we have now traced the problem to an incorrect specification of the specific heat capacity of air at low air temperatures, which we have now corrected. We have also imposed a mask over the Greenland ice sheet where the simulation of soil moisture does not make sense. The following Figure is the corrected version, which will be included in the revised manuscript.

*IC Prentice, on behalf of all authors*

---

## Author Response (AR2)

Dear Didier Roche;

Thank you very much for your thorough review of our manuscript. We understand that there needs to be a more comprehensive look at the SPLASH model performance. To accomplish this, we have taken your advice and ran the SPLASH model at six different climate regions at various elevations across North America (in order to retain comparability with the Livneh et al., 2015 datasets). This time we utilized monthly CRU TS3.23 data as an input to SPLASH (instead of mixing daily WATCH with monthly CRU) and allowed the model to equilibrate for one year (1991) and spin-up for eight years (1992–1999). The results are once again for a single year (2000).

Please find our updated manuscript, which has a completely re-written Section 4.1 regarding the local temporal trends and bioclimatic indices. A new figure has been added to show the six sites (Figure 3). Figures 4 and 5 (previous Figures 3 and 4) have been updated with the daily and monthly results of the six sites. Due to the increased amount of data, we have simplified the results shown in Figures 4 to highlight the main outputs (i.e., net radiation, soil moisture, and evapotranspiration) and condensed the monthly evapotranspiration to a single pane in Figure 5. For completeness, we created a supplemental PDF that includes the full time series plots (similar to the previous Figure 3) for each of the six climate sites. We have also noted in our discussions how some of the limitations and simplifications of the SPLASH model have influenced the new results. We hope that these new results and analyses will shed light on our model's performance and utility.

Sincerely,

Tyler Davis and co-workers

[revised manuscript text omitted]

---

## Author Response (AR4)

Dear Didier Roche,

Please find our updated manuscript with markups presented below. We have addressed the ambiguity of the color bars for the difference plots in Figures 6 and 7 with additional labels in the legend and expanded descriptions in the figure captions. Also, we made a minor note in Section 2.2 to the citation for the water-to-energy conversion factor.

Thank you again for your careful attention and assistance in guiding this work.

Sincerely,

Tyler Davis on behalf of all co-authors

[revised manuscript text omitted]